# Single cell RNA-sequencing identifies a metabolic aspect of apoptosis in *Rbf* mutant

Majd M. Ariss [1], Abul B.M.M.K. Islam [2], Meg Critcher[1], Maria Paula Zappia [1] & Maxim V. Frolov[1]

The function of Retinoblastoma tumor suppressor (pRB) is greatly influenced by the cellular context, therefore the consequences of pRB inactivation are cell-type-specific. Here we employ single cell RNA-sequencing (scRNA-seq) to profile the impact of an *Rbf* mutation during *Drosophila* eye development. First, we build a catalogue of 11,500 wild type eye disc cells containing major known cell types. We find a transcriptional switch occurring in differentiating photoreceptors at the time of axonogenesis. Next, we map a cell landscape of *Rbf* mutant and identify a mutant-specific cell population that shows intracellular acidification due to increase in glycolytic activity. Genetic experiments demonstrate that such metabolic changes, restricted to this unique *Rbf* mutant population, sensitize cells to apoptosis and define the pattern of cell death in *Rbf* mutant eye disc. Thus, these results illustrate how scRNA-seq can be applied to dissect mutant phenotypes.

[1] Department of Biochemistry and Molecular Genetics, University of Illinois at Chicago, 900 S Ashland Avenue, Chicago, IL 60607, USA. [2] Department of Genetic Engineering and Biotechnology, University of Dhaka, Dhaka 1000, Bangladesh. Correspondence and requests for materials should be addressed to M.V.F. (email: mfrolov@uic.edu)

Functional inactivation of the retinoblastoma protein (pRB) is considered an obligatory event in the development of human cancer and is usually attributed to its ability to block cell-cycle progression through negative regulation of the E2F transcription factor. Binding to pRB inhibits E2F transcriptional activity and halts cell cycle. Conversely, the inactivation of pRB releases E2F and allows S-phase entry[1]. Such a simplistic view is built on the assumption that pRB operates in the same way across different cell types. However, mouse models and clinical studies have revealed that the function of pRB is greatly influenced by the cellular context. The consequences of pRB inactivation are thought to be determined by a unique, cell-type-specific molecular circuitry around pRB. Such specific interactions may also help to explain why cancer originates in a specific cell type. For example, human retinoblastoma is believed to be derived from post-mitotic cone precursors. These cells are uniquely sensitive to Rb loss as they express cone lineage factors (TRβ2 and RXRγ) and the oncoproteins MYCN and MDM2[2]. Thus, it is important to understand how mutations in the RB pathway affect individual cell types. This point is especially relevant in interpreting the results of genome-wide studies, which have been extensively used to deduce how the RB pathway operates. However, averaging gene expression using bulk samples does not provide sufficient resolution to determine the impact of RB pathway mutations on individual cell types.

Recent advances in single-cell RNA-sequencing (scRNA-seq) offer an opportunity to detect variation at the cellular level and dissect heterogeneous tissues into unique cell clusters. Surprisingly, although scRNA-seq has been used to study tumor heterogeneity in cancer, this technology has yet to be adapted to dissect the mutant phenotypes in model organisms. *Drosophila* has a streamlined version of the mammalian RB pathway and proved to be invaluable in deciphering its role in vivo[3]. For example, investigating the mutant phenotype of *Rbf*, encoding the pRB ortholog, provided key insights into the function of *Rbf* in development. Like in mammalian cells, inactivation of *Rbf* in the larval eye imaginal disc results in mild cell-cycle defects and apoptosis. Increased sensitivity to apoptosis of *Rbf*-deficient cells is commonly attributed to elevated expression of apoptotic E2F target genes such as *hid* in flies. Notably, despite *hid* being upregulated throughout almost the entire *Rbf* mutant eye disc, apoptosis is restricted to cells anterior to the morphogenetic furrow that show a transient reduction in epidermal growth factor receptor (EGFR) signaling[4]. Thus, the *Rbf* mutant eye disc represents an ideal setting to apply scRNA-seq methodology and identify a precise cellular context that makes *Rbf* mutant cells sensitive to apoptosis.

Here, we report an atlas of 11,500 wild-type eye disc cells with 1× cellular coverage that includes major cell types in the developing larval eye. We also find a transcriptional switch during photoreceptor differentiation. We then utilize this resource to examine the *Rbf* mutant phenotype and identify a specific population of cells with increased glycolysis that makes them sensitive to E2F-dependent apoptosis. Thus, our results illustrate the applicability of scRNA-seq to profile mutant phenotypes.

## Results

**A cell atlas of the wild-type third-instar larval eye disc.** The *Drosophila* eye remains a preferable model to investigate the control and coordination of cell proliferation, differentiation and apoptosis. During the third-instar larval stage, the morphogenetic furrow (MF) sweeps across the eye disc from the posterior margin towards the anterior, demarcating the onset of neuronal differentiation. The asynchronously dividing cells of the anterior compartment arrest at G1 upon entry into the MF, specifying into photoreceptors as they exit posteriorly. However, not all cells commit to the differentiation program upon emerging from the MF. The remaining uncommitted interommatidial cells undergo one synchronous round of cell division called the second mitotic wave (SMW) posterior to the MF and then remain quiescent (Fig. 1a).

There are several transcriptional domains in the eye disc. The pre-proneural (PPN) domain located immediately anterior to the MF is defined by the expression of *hairy* (*h*) and *dachshund* (*dac*) (Fig. 1a, b)[5]. Cells anterior to the PPN express *homothorax* (*hth*) in the eye-antennal domain as well as the apical peripodial epithelium, a thin squamous membrane overlying the eye disc proper, which gives rise to the head capsule (Fig. 1a, b). In the posterior compartment, photoreceptors are organized in repetitive units called ommatidia. Eight photoreceptors (R1–R8) are recruited into each ommatidium in a sequential manner and can be distinguished by specific markers: Senseless (Sens) marks R8 (Fig. 1b), BarH1/2 are expressed in R1 and R6, while Rough (Ro) delineates R2 and R5[6]. Following specification, photoreceptors project their axons in the basal compartment to form connections with migratory glial cells from the brain.

We began by building a catalog of cells in the wild-type third-instar larval eye disc using Drop-seq, a microfluidic-based scRNA-seq platform[7]. The third-instar larval eye discs were dissected at the optic stalk and eye-antennal domain, then dissociated into a single-cell suspension. To control for potential batch effects, a total of 550 eye discs were dissected over 11 biological replicates and scRNA-seq was performed on each replicate. Following quality controls, sequencing data were collected from 11,416 individual cells at an average depth of 30,344 mapped reads per cell, with a total of 16,331 genes detected. To unbiasedly classify cell types, we performed a t-stochastic neighbor embedding (tSNE) analysis using Seurat[8] and identified 15 distinct cell clusters (Fig. 1c, Supplementary Table 1). Cells from each of 11 biological replicates were evenly distributed among the cell populations and contributed to each cluster, with the exception of the rare cell-type HEMO shared by 10 replicates (Supplementary Fig. 1a). This demonstrates that the results are highly reproducible and show no apparent batch effect between samples.

To assess the efficiency of cell dissociation in generating a single-cell suspension, we took advantage of known markers that are highly specific for individual photoreceptors. Sens is expressed only in R8 and therefore its expression is mutually exclusive with Rough, a marker for R2/5, or BarH2, a marker for R1/6. Since R8 is positioned in the center of each ommatidium, it is making physical contact with other photoreceptors. If tissues were not effectively dissociated, we expected to find co-expression of Sens with other photoreceptor-specific transcripts such as Rough or BarH2 in single-cell libraries. Significantly, less than 4% of libraries contained mixed R cell-specific transcripts (i.e., *sens* and *ro* transcripts present in the same library), suggesting that >95% represent single-cell transcriptomes (Fig. 1d). We calculated the expected co-expression values based on independent expression of the three genes and note that our observed results (1.7% and 3.6%) (Fig. 1d) are significantly lower than expected values ($p$ value = $2.3 \times 10^{-7}$ using chi-squared test).

Unsupervised Seurat clustering identified 15 cell populations each having a set of differential expressed genes (Supplementary Table 1, Supplementary Data 1 and Fig. 2). Each population was assigned based on known markers computationally derived from the analysis (Fig. 3a, Supplementary Table 1). The MF was distinguished by high levels of Notch signaling (Fig. 1b) including detection of several *E(spl)* genes (Fig. 3a)[9]. The expression of *h* and *dac* was used to identify the PPN and, consistent with previous literature, the latter was also observed in the MF (Fig. 3a)[5].

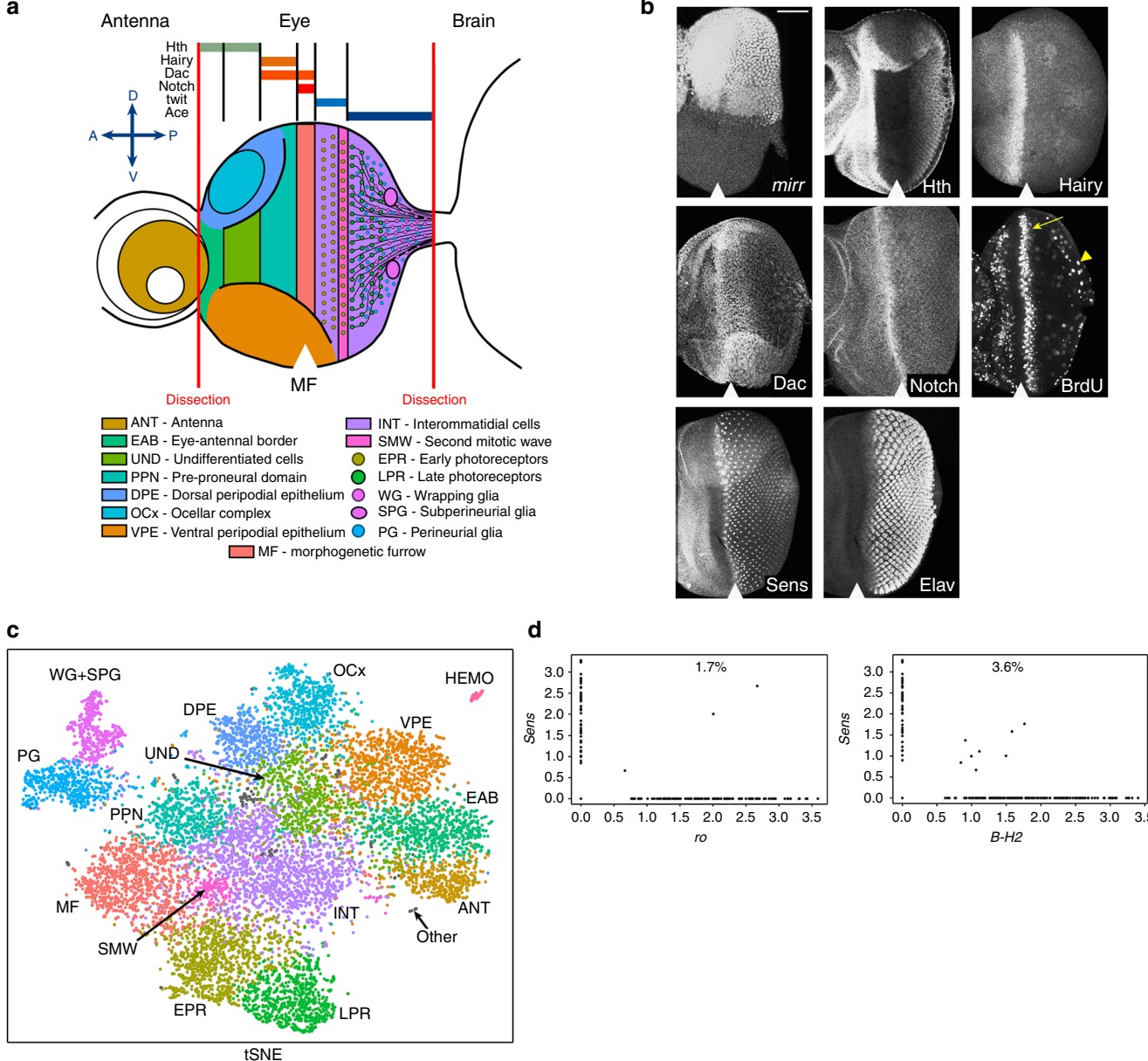

**Fig. 1** A cell atlas of the wild-type third-instar larval eye disc. **a** A cartoon representation of the eye-antennal disc and known cell types that are color coded. The red vertical lines indicate where the dissections were made. The expression domains of genes above that were used to assign cell clusters. D dorsal, V ventral, P posterior to the MF, A anterior to the MF. **b** The pattern of the expression of cell-type markers in the eye as revealed by immunofluorescence. BrdU labeling shows the location of SMW (yellow arrow) and PG (yellow arrowhead) cells undergoing cell division. The scale bar is 50 μm. **c** The tSNE representation of 11,416 wild-type eye disc cells. Each cluster represents a cell type/domain, characterized by a unique gene expression signature. Clusters are color coded as in (**a**). Other cells are labeled as gray. **d** Gene/gene plots generated on all photoreceptors for *sens* (R8) and *ro* (R2/5), *sens* and *B-H2* (R1/6). The percentage represents doublets, based on the detection of two R cell-specific markers in the same single-cell library. The position of the MF is shown by white arrowhead

Undifferentiated cells (UND) located anterior to the PPN were identified by the presence of *hth* and *toy* expression.

The eye-antennal border (EAB) forms the boundary between these two tissues. EAB cells were assigned by the expression of *Lim1* and *cut*[10] (Fig. 3a). Since the dissection included part of the antenna (ANT), these cells were distinguished by the expression of *Dll* (Fig. 3a)[11].

Four cell populations were mapped to the posterior compartment of the eye disc. Two of which (early photoreceptor (EPR) and late photoreceptor (LPR)) were identified as photoreceptors based on the presence of the classical neuronal markers *elav*, *futsch* and *Appl* (Fig. 3a). The remaining two are interommatidial

cells (INT and SMW) since they expressed posterior markers such as *ed* (Supplementary Table 1, Supplementary Data 1) and *Mmp2* (Fig. 3a) but lacked the expression of neuronal genes. As described above, interommatidial cells undergo one synchronous round of cell division in the second mitotic wave and exit the cell cycle in the posterior (Fig. 1b)[6]. Since cells in the SMW cluster exhibit high expression of cell-cycle genes such as *Claspin*, *PCNA*, *Mcm7* (Fig. 3b), *dap* and *stg* (Supplementary Table 1, Supplementary Data 1), we concluded that these are cells of the second mitotic wave. Conversely, the expression of cell-cycle genes is low in INT cells and therefore they are the quiescent interommatidial cells. Interestingly, we did not find a distinct cell population

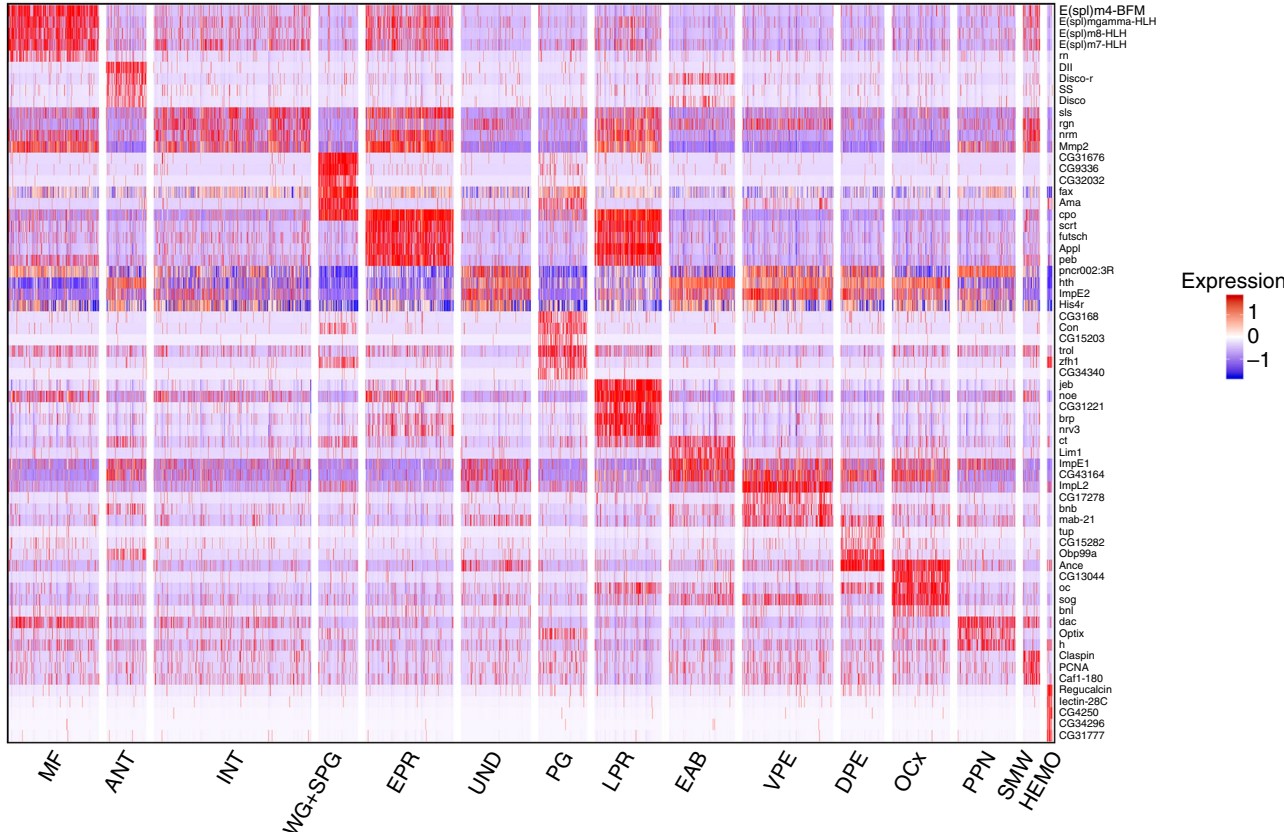

**Fig. 2** A heat map showing markers across cell populations. Heat map displaying the expression of 65 top markers across 15 wild-type populations

corresponding to cone cells but their respective marker *cut* was detected throughout the LPR population. This is likely because cone precursors share similarities with photoreceptors and therefore Seurat analysis does not distinguish between them at this developmental stage. We identified several cell type-specific markers for the eye disc. Neuromusculin (encoded by *nrm*) was previously shown to be only expressed in photoreceptors[12]. However, *nrm* expression was also detected in interommatidial cells in our data set and this was validated by immunofluorescence (Fig. 3a, c). Interestingly, computational analysis identified a muscle-specific gene *sls* (also known as *kettin*)[13] as a posterior marker (Fig. 3c). We confirmed this surprising observation by using a *sls* reporter that showed robust expression in the posterior domain of the eye disc (Fig. 3c).

The tSNE analysis revealed two glial cell populations (PG and WG+SPG) based on *repo* expression (Fig. 3a, Supplementary Data 2), a well-established glial marker[6]. These clusters are distinct by their proliferation and differentiation states. Perineurial glia (PG) are undifferentiated, dividing cells that label with 5-bromo-2'-deoxyuridine (BrdU)[14] and can be detected by immunofluorescence in the posterior compartment of the eye disc (Fig. 1b). Since PG cell population exhibited high expression of cell-cycle genes *Claspin*, *stg* and *PCNA* (Fig. 3b), we assigned this population to perineural glial cells. Consistently, the PG-specific gene *CG3168*[15] was also exclusively detected in this population (Fig. 3a). As PG cells migrate towards the eye and make contacts with photoreceptor axons, they exit cell cycle and differentiate into wrapping glia (WG). These two types of glial cells are physically separated by two large subperineurial, or carpet, glial cells (SPG). The presence of the known WG markers *Gli*, *nrv2* and *sty* (Supplementary Table 1, Supplementary Data 1) and the SPG marker *moody*[16] (Fig. 3a) was used to assign the WG+SPG cluster to wrapping glia and subperineural glial cells.

Consistently, the expression of cell-cycle genes was low in these cells (Fig. 3b) as they are no longer proliferating. Additionally, we identified two markers of wrapping glial cells, *NK7.1* (most highly expressed in WG+SPG) and *CG9336*, expressed exclusively in this cluster and was previously presumed to be expressed in axon projections[17]. The glial-specific expression of *NK7.1* and *CG9336* was confirmed by a green fluorescent protein (GFP) reporter and fluorescence in situ hybridization (FISH), respectively (Fig. 3d). Another gene is *couch potato* (*cpo*) that has been previously shown to be expressed throughout the posterior compartment[18]. We found that its expression is restricted to photoreceptors and absent in interommatidial cells (Fig. 3a, c). *cpo* is strongly expressed in wrapping glial cells of WG+SPG (Fig. 3d) and this was validated with a *cpo-lacZ* reporter line (Fig. 3b, d).

We discerned two cell clusters representing cells of the peripodial epithelium, a thin squamous membrane that overlies the eye disc proper and gives rise to head structures during metamorphosis. This membrane is composed of two distinct cell populations; the dorsal peripodial epithelium (DPE), and the ventral peripodial epithelium (VPE). These populations were distinguished by dorsal markers such as *mirror* (Fig. 1b)[19] that is present in DPE but absent in VPE (Supplementary Table 1, Supplementary Data 1). The VPE expresses *cv-c*, a top marker for the EAB[20]. However, the VPE does not express *Lim1* which further distinguishes that population from the EAB. Cells of the ocellar complex (OCx), which form the ocelli on the adult head, were identified by the expression of *ocelliless* (*oc*)[21]. Finally, we identified a cluster of hemocytes (HEMO) consisting of around 60 cells (0.53% of all cells in our dataset), based on the expression of the classical marker *Hemolectin* (*Hml*)[22] among others (Fig. 3a and Supplementary Table 1, Supplementary Data 1). From our computational analysis we identified *pigs*, a gene that plays a role in ovary and muscle development[23], as a hemocyte marker and

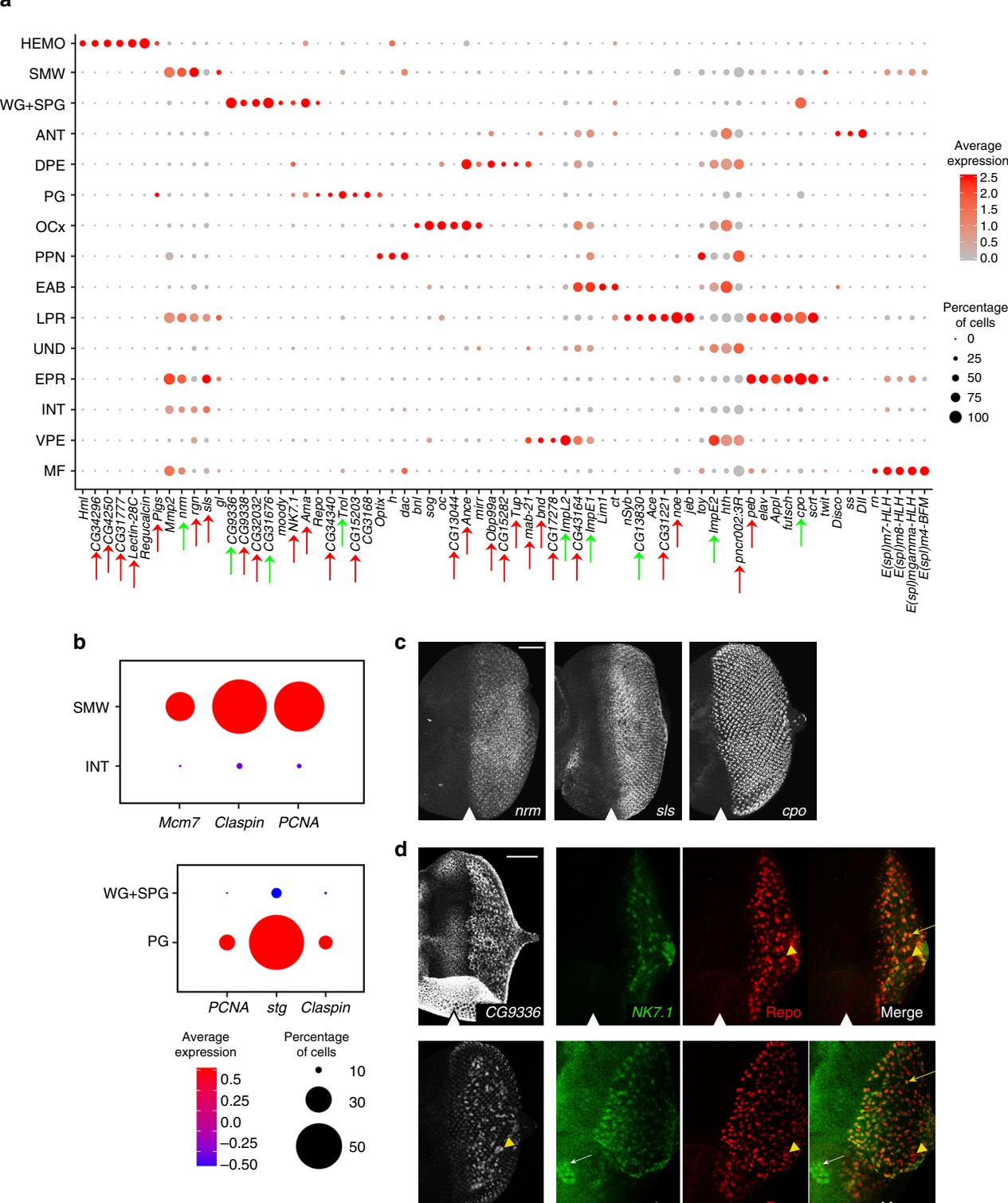

**Fig. 3** Single-cell RNA-seq identifies cell-specific markers. **a** Dot plot of top markers for all populations. Red arrows point at previously unreported eye cell-type markers. Green arrows indicate genes previously studied in the eye that were not assigned to a specific cell type. The genes without arrows are known markers used to assign cell populations. **b** LacZ enhancer traps for posterior candidate markers *neuromusculin* (*nrm*), *sallimus* (*sls*, also known as *kettin*) and photoreceptor marker *couch potato* (*cpo*). **c** Dot plots, top panel shows interommatidial populations INT and SMW, showing expression of cell-cycle genes exclusively in SMW subpopulation. Bottom panel shows glial cell populations PG and WG+SPG showing expression of cell-cycle genes exclusively in PG. The scale bar is 50 μm. **d** Wrapping glia markers *CG9336* and *NK7.1* of and their expression as revealed by FISH (*CG9336*) and a GFP reporter for *NK7.1* (green). Repo (red) is a glial marker. Merge image marks cells co-expressing *NK7.1* and Repo, also shown with yellow arrow. Markers of PG, *cpo* and *pigs* are revealed by LacZ enhancer trap and GFP reporter, respectively. SPG are indicated by a yellow arrowhead, overlap of Repo and *pigs* shown with yellow arrow. The scale bar is 50 μm. The position of the MF is shown by white arrowhead

confirmed its expression in hemocytes using a *pigs* reporter (Fig. 3a, d). In addition to hemocytes, *pigs* is expressed in PG cells as revealed by Seurat and immunofluorescence (Fig. 3a, d). The identification of a distinct HEMO cluster within the 10,000 cells of the eye disc[24] is notable as only a small number of hemocytes are associated with each disc. We concluded that our atlas of 11,415 cells accurately represents known cell types of the larval eye disc and efficiently captures rare cell types (<1%). Additionally, our results identified unique transcriptional signatures within glial cells and two distinct photoreceptors populations.

**Developing photoreceptors undergo a transcriptional switch**. Given that there are eight photoreceptors, the tSNE analysis unexpectedly detected only two photoreceptor clusters, EPR and LPR. To determine whether EPR and LPR represent photoreceptors of different R types, we plotted the R-cell-specific markers *sens*, *ro* and *B-H2* against *elav*, a pan-neuronal marker expressed in all R cells. Surprisingly, cells expressing *sens*, *ro* and *B-H2* were found in both clusters (Fig. 4a) revealing lack of bias towards a particular photoreceptor type between EPR and LPR populations. In order to determine whether there is an R-type segregation within each cluster, we selected the cells in EPR and performed a supervised Seurat analysis on these cells only. Yet, *sens-*, *ro-* and *B-H2*-positive cells remained randomly scattered throughout the tSNE feature plots, revealing lack of clustering based on R-type (Supplementary Fig. 1b). Similar result was observed when Seurat analysis was done only on LPR cells (Supplementary Fig. 1b).

The comparison of top marker gene lists for EPR and LPR revealed a significant overlap, as the majority of EPR markers were also found in LPR (Fig. 4b). One of the rare EPR-specific markers is *twit*. We used *twit* probe for FISH to localize the position of EPR in the eye disc and detected its expression within a narrow band posterior to the MF, ending at column 4 (Fig. 4c, d). Conversely, the expression of the LPR-specific top marker *Ace*, encoding the enzyme Acetylcholine esterase that degrades acetylcholine in synapses, was detected posterior to column 4, and in photoreceptor axons in the basal compartment (Fig. 4c, d). Since 153 genes were upregulated in LPR in comparison to EPR (Fig. 4b), we decided to further elucidate the differences between EPR and LPR. We performed gene set enrichment analysis for gene ontology biological processes (GOBP) on all computationally derived genes for EPR, LPR and UND. As expected, undifferentiated cells were dominated by cell cycle-related categories. In contrast, EPR and LPR showed enrichment for photoreceptor differentiation-related GO terms (Fig. 4e and Supplementary Data 3–5). Interestingly, LPR genes exhibited a statistically significant enrichment of axonogenesis and related categories compared to EPR. Accordingly, computationally identified top markers for LPR *jeb*, *brp*, *Dscam2* and *nSyb* have a known role in axon guidance[25–28].

Finally, we employed Monocle 2[29] to determine the temporal order of development of EPR and LPR along the pseudotime-derived neuronal differentiation axis. The progression of cellular differentiation was visualized on the plot; from UND progenitor cells to PPN in the anterior, through MF to INT in the posterior, and finally to EPR and LPR (Fig. 4f and Supplementary Fig. 2). Notably, LPR clustered consecutively to EPR in the plot, with the latter located closer to MF cells. We concluded that photoreceptors cluster irrespective of R-cell type. However, after column 4, photoreceptors initiate axonogenesis[6] and this transcriptional switch distinguishes late (LPR) from early (EPR) photoreceptors.

**scRNA-seq identified an *Rbf* mutant-specific cell cluster**. Having built a cell atlas of the wild-type third-instar larval eye disc, we proceeded to investigate the effect of an *Rbf* mutation on individual cell types in this tissue. We performed scRNA-seq on *Rbf120a* mutant eye discs[4] in 3 replicates and generated a scRNA-seq dataset of 5203 *Rbf* mutant cells. We then selected 5591 wild-type cells and conducted a tSNE analysis on the combined dataset containing both wild-type and *Rbf* mutant cells.

The tSNE plot identified 15 distinct cell clusters (Fig. 5a) that were readily assigned based on markers from the wild-type cell atlas (Fig. 1a, Supplementary Table 1, Supplementary Data 1). Derepression of cell-cycle genes is a known hallmark of *Rbf* mutant[4]. Accordingly, *PCNA*, *Mcm7* and others cell-cycle genes were upregulated in *Rbf* mutant cells (Fig. 5b). Interestingly, Seurat analysis no longer groups cells of the SMW into a distinct cluster, likely because SMW cells are highly proliferative and therefore become indistinguishable from other *Rbf* mutant cells. tSNE analysis revealed that cells of both genotypes contributed to each cell population, albeit with small variations between the numbers of wild-type and *Rbf* mutant cells in several cell populations. For example, the number of photoreceptors in *Rbf* mutant appeared to be lower than in the wild type. However, these differences are likely to be due to a subtle developmental delay of *Rbf* mutants that affects the position of the morphogenetic furrow at the time of dissection and therefore influences the number of photoreceptors that are recruited in the posterior.

In striking contrast to subtle differences described above, cluster 13 consisted of almost exclusively *Rbf120a* cells (Fig. 5a). The *Ldh* (*ImpL3*) gene encoding lactate dehydrogenase is the top ranked marker and its expression is restricted to cells of that cluster (Fig. 5b, c and Supplementary Data 6–7). Therefore, to identify the spatial location of cluster 13 in the eye disc, we examined the pattern of *Ldh* expression in *Rbf* mutant discs by in situ hybridization. Notably, *Ldh* was highly expressed in a distinct stripe preceding the MF in *Rbf* mutant eye discs, while being undetectable in wild-type eye discs (Fig. 5d). This is consistent with previous observations of very low endogenous *Ldh* activity at this developmental stage[30]. Two other predicted markers for cluster 13 were *Ald*, encoding the glycolytic enzyme Aldolase, and *HIF1a* (*sima*), a transcription factor that directly regulates *Ldh* in flies and mammals (Fig. 5c and Supplementary Data 6–7)[30, 31]. Consistently, both *Ald* and *HIF1a* matched the pattern of *Ldh* expression (Fig. 5d).

One consequence of upregulated glycolytic gene expression is increased glycolytic flux and subsequently excessive production of lactate which decreases intracellular pH. To determine whether this phenomenon occurs in *Rbf120a* mutant, we used a live intracellular probe that fluoresces only in low pH conditions. As expected, no signal was detected in control eye discs (Fig. 5e) as the expression of glycolytic genes is low during late larval development[32]. Strikingly, *Rbf* mutant exhibited a high level of intracellular acidification immediately anterior to the MF (Fig. 5e) that largely corresponds to the region of elevated expression of *Ldh* and *Ald* (Fig. 5d). This is due to increased glycolytic activity as downregulation of these genes by RNA interference (RNAi) in the *Rbf120a* background completely suppresses intracellular acidification (Fig. 5e). We concluded that an *Rbf* mutation leads to upregulation of glycolytic gene expression that is restricted to a population of cells anterior to the MF. This consequently lowers intracellular pH through increased glycolytic activity.

**Intracellular acidification increases apoptosis in *Rbf120a***. Loss of *Rbf* results in upregulation of the apoptotic gene *hid* that sensitizes cells to apoptosis. Curiously, cell death does not occur throughout the entire *Rbf* mutant eye disc. Instead, apoptosis is restricted to a stripe of cells immediately anterior to the MF (Fig. 6a)[4], which was attributed to a transient reduction in pro-

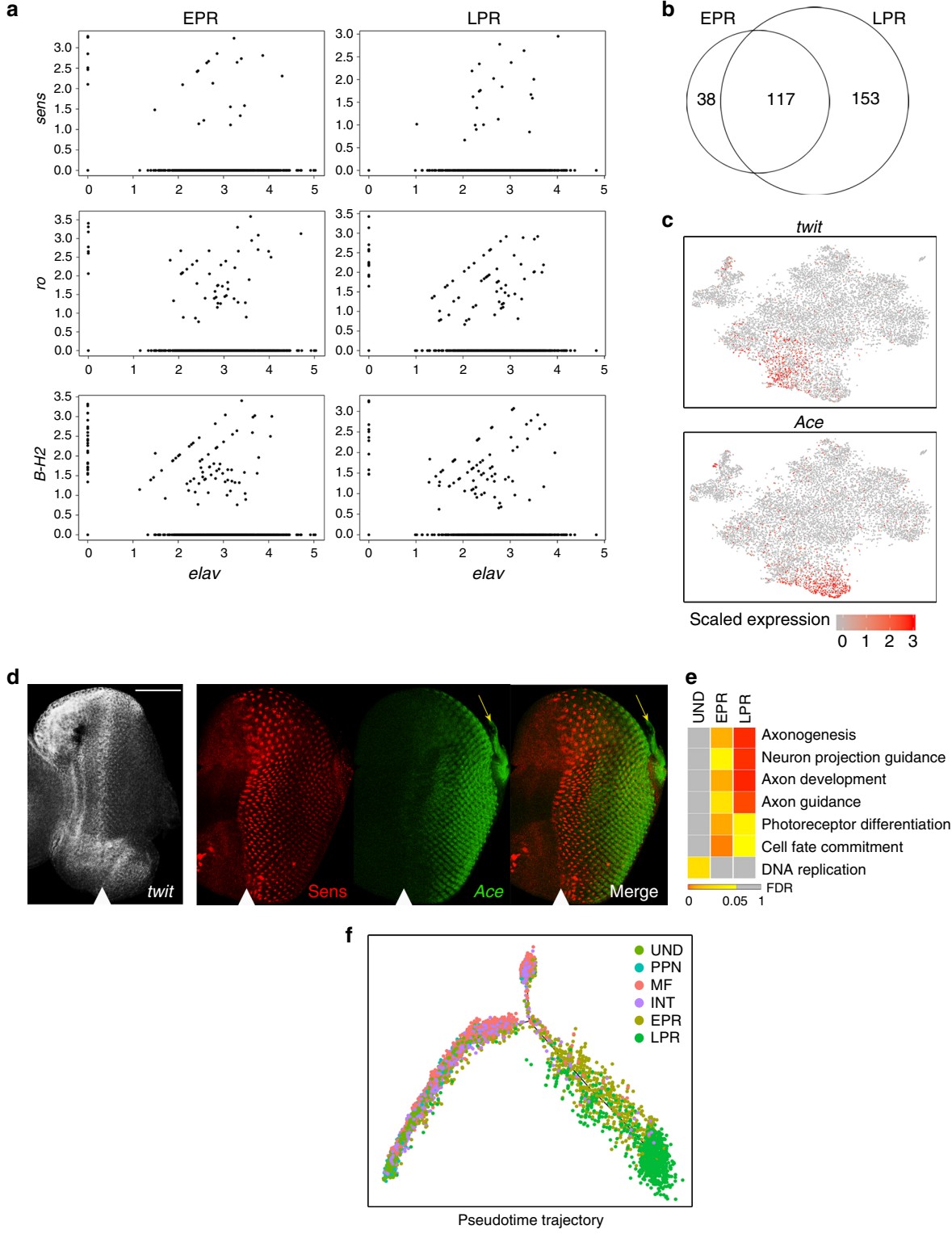

**Fig. 4** Photoreceptors undergo a transcriptional switch at onset of axonogenesis. **a** Gene/gene plots showing expression of R type-specific markers *sens*, *ro* and *B-H2* in relation to *elav* in EPR (left panel) and LPR (right panel). **b** Venn diagram of the gene lists for EPR and LPR clusters. **c** Feature plots on tSNE for *twit* and *Ace* show predominant expression of each gene in EPR and LPR cells, respectively. **d** EPR and LPR marker gene localization. Left panel: FISH for *twit* mRNA, an EPR-specific gene, shows expression in the MF and posterior until column 4. Right panel: GFP enhancer trap for *Ace* (green), an LPR top marker, was costained with Sens (red). *Ace-GFP* was detected after column 4 and in axonal projections (yellow arrow). The scale bar is 50 μm. **e** Gene enrichment analysis for biological processes in UND, EPR and LPR. **f** Monocle 2 trajectory using shown clusters. Stages of differentiation are visualized, with the split of INT and photoreceptor populations. LPR arise from the EPR population, clustering consecutively in pseudotime. The position of the MF is shown by white arrowhead

survival EGFR signaling in this region. We noted that the pattern of apoptosis appears to match the location of the *Rbf* mutant-specific cluster 13 that also showed the expression of *hid* as a top marker (Fig. 5c, d and Supplementary Data 6–7). Since

upregulation of *Ald* and *Ldh* and increased intracellular acidification are the major features of this cluster, we examined their contribution to apoptosis in *Rbf* mutant. RNAi was used to downregulate expression of each *Ald*, *Ldh* and *HIF1a* in the

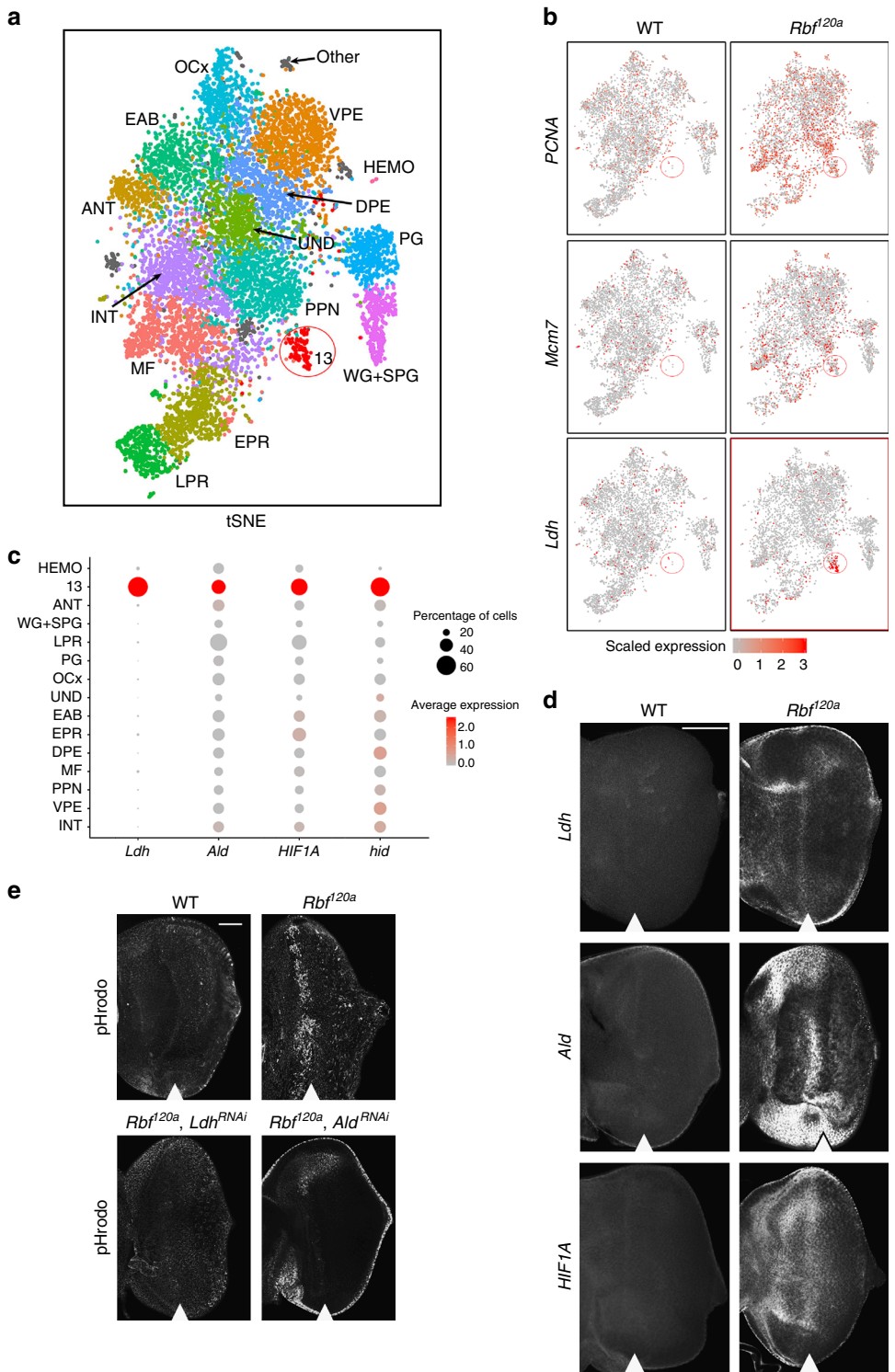

**Fig. 5** Wild-type and *Rbf^{120a}* scRNA-seq identifies a mutant-specific metabolic cluster. **a** Seurat analysis using a combined wild-type and *Rbf* mutant single-cell transcriptomes. Cluster 13, outlined in red, is almost exclusively comprised of *Rbf^{120a}* mutant cells. Clusters are color coded as in Fig. 1a. **b** Feature plots on tSNE showing the expression of *PCNA*, *Mcm7* and *Ldh* in wild-type and *Rbf* mutant cells. Location of cluster 13 is outlined in red. **c** Dot plot showing expression of cluster 13 markers *Ldh*, *Ald*, *HIF1a* and *hid* throughout all clusters. **d** Expression of *Ldh*, *Ald* and *HIF1A* in the eye disc as revealed by FISH. The scale bar is 50 μm. **e** Pattern of intracellular acidification as revealed by pHrodo. The scale bar is 50 μm. The position of the MF is shown by white arrowhead

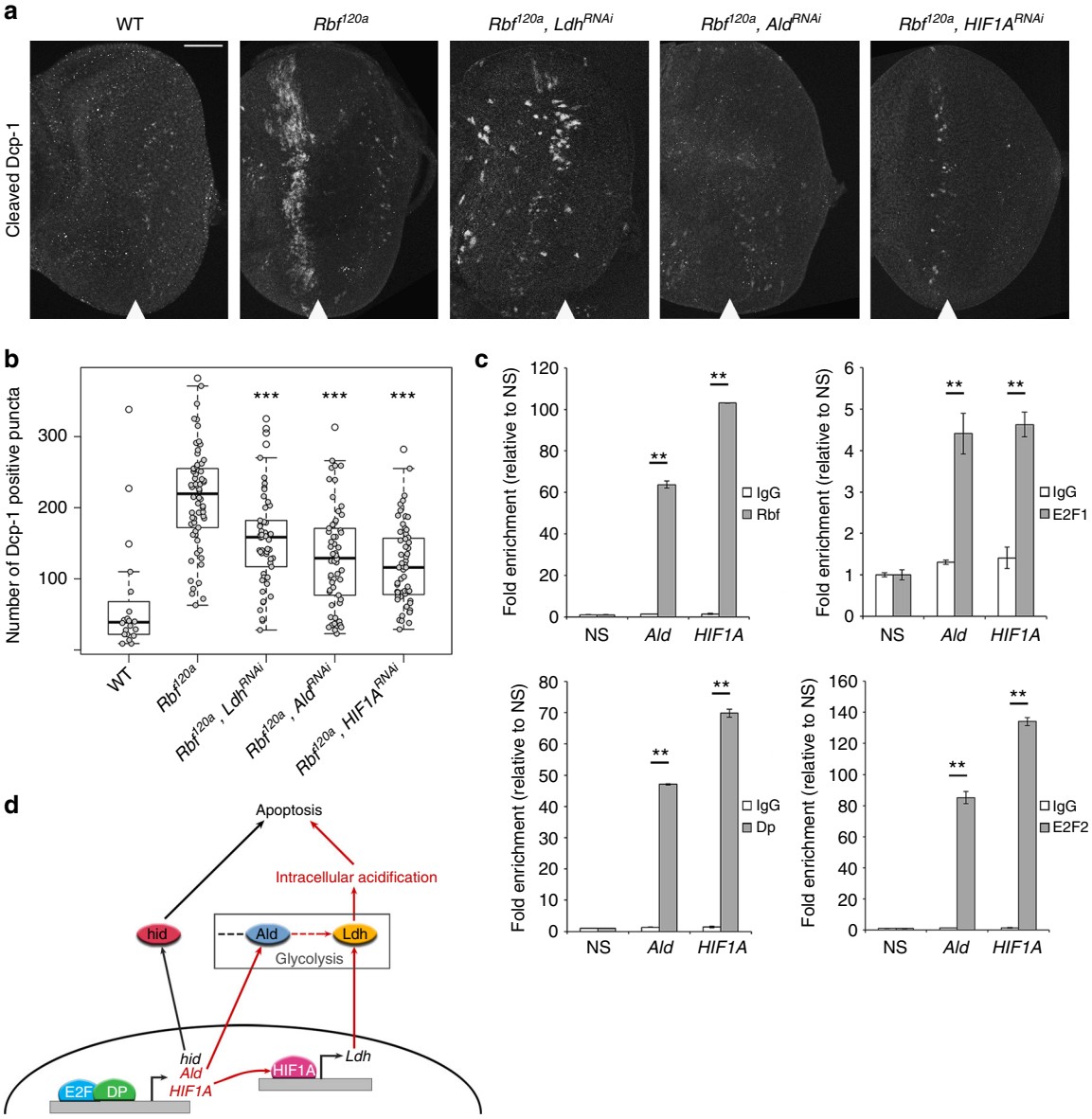

**Fig. 6** Upregulation of *Ldh*, *Ald* and *HIF1a* make *Rbf* mutant prone to apoptosis. **a** Cleaved Dcp-1 immunofluorescence displays the distinct wave of apoptosis in *Rbf[120a]* that is rescued by downregulation of each *Ldh*, *Ald* and *HIF1a*. The scale bar is 50 μm. **b** Quantification of cleaved Dcp-1-positive puncta in N (WT) = 20 eyes, N (*Rbf[120a]*) = 62 eyes, N (*Rbf[120a]*, *Ldh[RNAi]*) = 54 eyes, N (*Rbf[120a]*, *Ald[RNAi]*) = 58 eyes, and N (*Rbf[120a]*, *HIF1A[RNAi]*) = 59 eyes. Center box line represents the median. The lower box limit is the first quartile. The upper box limit is the third quartile. The whiskers point at the variabilities outside the first and third quartiles. Values outside the whiskers are outliers. The hollow circles are outliers. One-way analysis of variance (ANOVA) test compares knockdowns versus *Rbf[120a]*, \*\*\*adjusted *p* value < 5 × 10[-4]. **c** ChIP-qPCR performed on larval chromatin using Rbf, E2f1, E2f2 and Dp antibodies at the promoter regions of *Ald* and *HIF1a* relative to the negative site (NS). Data are representative of two experimental replicates and show the mean and error bars represent s.e.m., two-way ANOVA test was used, \*\**p* value < 0.001. **d** Molecular mechanism of the metabolic aspect of apoptosis in *Rbf* mutant cells. The position of the MF is shown by white arrowhead

*Rbf[120a]* background and apoptotic cells were visualized by immunostaining for cleaved Dcp-1, the *Drosophila* homolog of effector Caspase-7. Strikingly, the downregulation of *Ldh*, *Ald* or *HIF1a* significantly suppressed cell death in *Rbf* mutant cells (Fig. 6a, b and Supplementary Table 2). Collectively, these results support the interpretation that cluster 13 consists of *Rbf* mutant cells undergoing apoptosis. Importantly, intracellular acidification is not merely a consequence of apoptosis in *Rbf* mutants, since pH remains normal when cell death is induced by overexpression of *rpr* or *hid* (Supplementary Fig. 3). These results indicate that increased intracellular acidification occurs independently of the activation of pro-apoptotic signals in vivo.

In order to understand the regulation of *Ldh*, *Ald* and *HIF1a* in the *Rbf* mutant, we isolated chromatin from whole wild-type third-instar larvae and performed chromatin immunoprecipitation–quantitative PCR (ChIP-qPCR). Compared to a nonspecific antibody (immunoglobulin G (IgG)), the promoter regions of *Ald* and *HIF1a* were significantly enriched with Rbf, Dp, E2f1 and E2f2 (*p* value < 0.001 using a two-way analysis of variance (ANOVA) test) (Fig. 6c and Supplementary Table 3). Since *HIF1a* is known to directly activate *Ldh*[30], its upregulation in *Rbf* mutant is likely to be mediated by hypoxia-inducible factor-1α (HIF1α). These findings indicate that Rbf directly regulates the expression of *Ald* and *HIF1a* to prevent their inappropriate activation that leads to apoptosis (Fig. 6d).

## Discussion

Here, we employed scRNA-seq to characterize a mutant phenotype. tSNE analysis on a combined scRNA-seq dataset of wild-type and *Rbf* mutant eye disc cells identified a small cluster that corresponds to *Rbf* mutant cells undergoing apoptosis. These cells exhibit elevated glycolytic activity and increased intracellular acidification that makes them uniquely sensitive to apoptosis. The results of our computational analysis were confirmed by genetic experiments. Thus, dissecting a mutant phenotype by scRNA-seq proved to be informative in unraveling acute cellular alterations that would remain undetected using conventional methods, such as whole tissue RNA-seq.

The proper assignment of computationally predicted clusters of cells inferred from scRNA-seq data to a particular cell type is dependent on prior information about cell type-specific markers. We therefore chose the *Drosophila* eye disc due to exceptional knowledge of the underlying biology. Our wild-type catalog contains ~11,500 cells and is comparable with the estimated count of 10,000 cells in the eye disc[24]. This 1× cellular coverage proved sufficient to identify known domains, cell types and states. For example, progenitor cells in the anterior compartment were split into two clusters that matched previously described transcriptional domains[5]. Notably, additional markers were identified for each cell population (Fig. 3a) and were often ranked higher than classical markers by computational analysis (Supplementary Data 1), raising the question of their importance in eye development.

Unexpectedly, photoreceptors did not segregate according to photoreceptor type (R1–8), which is usually defined by one or two markers. One possibility is that the 1× cellular coverage, or relatively low gene capture in Drop-seq, may be insufficient to detect transcriptional variations between R types. Alternatively, this may indicate that transcriptional variations between R types are minimal at the larval stage as photoreceptors are yet to express specific rhodopsins. Moreover, the type of rhodopsin (rh) does not correlate with R type. R1 through R6 express the broad-spectrum *rh1*, while R7 and R8 express one of the four rhodopsins, *rh3, rh4, rh5* or *rh6*, responsible for color vision[33]. Instead, we find that large transcriptional changes occur later in photoreceptor development, at the onset of axonal projections. These axons establish the crucial connections with glial cells migrating from the brain required for photoreceptor survival[34]. Thus, scRNA-seq identified a distinct transcriptional switch occurring in developing photoreceptors, between EPR and LPR, that is important for their survival. This also illustrates a potential limitation when interpreting single-cell expression data, as previous findings may provide a biased expectation of how cells should cluster.

Finally, our results provide a another perspective as to why the loss of *Rbf* sensitizes cells to apoptosis. Previous studies suggested that in *Rbf* mutant larval eye discs, E2F-dependent upregulation of *hid* and the transient reduction of pro-survival EGFR signaling leads to apoptosis anterior to the MF[4]. We show that increased intracellular acidification is another key event to trigger apoptosis exclusively in this domain. Whether intracellular acidification is a cause or consequence of apoptosis remains a long-standing conundrum. The idea that cellular acidification precedes cell death was initially shown in neutrophils[35]. Our results support and expand on this hypothesis by demonstrating the causative role of intracellular acidification in apoptosis in vivo. Since lactate is readily secreted, it is possible that glycolytic cells of population 13 induce apoptosis in adjacent cells. Although we cannot completely exclude this interpretation, we note that *hid* is highly expressed in cells of population 13 indicating that apoptosis occurs in a cell-autonomous manner. Importantly, pH remains neutral in cells undergoing *rpr*- and *hid*-induced cell death,

indicating that intracellular acidification is not merely a consequence of apoptosis. Whether these effects are specific to *Rbf* mutants is uncertain as it remains to be determined if elevated expression of glycolytic genes is sufficient to induce cell death in the wild-type discs. Nevertheless, our data suggest a simple model wherein loss of *Rbf* results in upregulation of the glycolytic genes such as *Ldh* and *Ald* immediately anterior to the MF. This in turn increases lactate production and lowers pH in these cells which promotes apoptosis (Fig. 6d). Although E2F directly elevates *Ald* expression, upregulation of *Ldh* is indirect and likely mediated by *HIF1a*, an E2F target. These findings underscore the sensitivity of scRNA-seq to detect transcriptional changes in a small cell population, which were previously missed in microarray and RNA-seq experiments using bulk samples.

Our work also highlights the value of a high-resolution wild-type cell atlas in mapping mutant scRNA-seq datasets. We and others[7, 36] note remarkable reproducibility of scRNA-seq among samples, as cells from almost a dozen replicates unbiasedly contributed to each population. We suggest that using single-cell genomics to profile mutant phenotypes may identify previously unappreciated biology by detecting restricted cellular perturbations.

## Methods

**Fly stocks**. All stocks and crosses were maintained at 25 °C in vials containing standard cornmeal agar medium.
    Wild-type stock used for scRNA-seq and genetic experiments: *y, v; attp2* (Bloomington#36303)
*Rbf[120a]/FM7, GFP* (gift from Nick Dyson)
*Mi{y[+mDint2]=MIC}NK7.1[MI02850]* (BDSC),
*Mi{PT-GFSTF.1}Ace[MI07345-GFSTF.1]* (BDSC),
*P{ry[+t7.2]=A92}nrm[A37] sr[1] e[s] ca[1]* (BDSC),
*P{w[+mC]=lacW}mirr[B1-12]/TM6B, P{w[+mC]=tub-QS.P}4A Tb[1]* (BDSC),
*P{w[+mC]=lacW}sls[j1D7]* (BDSC)
*P{ry[+t7.2]=lArB}cpo[2] ry[506]*(BDSC)
*Mi{y[+mDint2]=MIC}pigs[MI11007]* (BDSC)
*Rbf[120a]* ey-FLP / Y; act5C>FRT>stop>FRT>GAL4, UAS-GFP (gift from Nam-Sung Moon),
*P{y[+t7.7] v[+t1.8]=TRiP.JF02071}attP2* (BDSC),
*P{y[+t7.7] v[+t1.8]=TRiP.HMC06145}attP40* (BDSC),
*UAS-Ldh[RNAi]* (VDRC-KK#110190),
*P{y[+t7.7] v[+t1.8]=TRiP.HMS00832}attP2* (BDSC),
*P{y[+t7.7] v[+t1.8]=TRiP.HMS00833}attP2* (BDSC),
*P{w[+mC]=GAL4-ninaE.GMR}12* (BDSC),
*P{w[+mC]=GMR-rpr.H}S* (BDSC),
*P{w[+mC]=GMR-hid}SS1, y[1] w[*] P{ry[+t7.2]=neoFRT}19A; P{w[+m*]=GAL4-ey.H}SS5, P{w[+mC]=UAS-FLP.D}JD2* (BDSC)
*P{y[+t7.7] v[+t1.8]=VALIUM20-EGFP.shRNA.3}attP40* (BDSC)
Final genotypes:
*Rbf[120a]*, ey-FLP / Y; act5C>FRT>stop>FRT>GAL4 / UAS-Ald[RNAi]
*Rbf[120a]*, ey-FLP / Y; act5C>FRT>stop>FRT>GAL4 / UAS-Ldh[RNAi]
*Rbf[120a]*, ey-FLP / Y; act5C>FRT>stop>FRT>GAL4 / UAS-sima[RNAi]
*GMR-GAL4 / UAS-EGFP[RNAi]*.

**Fluorescent in situ hybridization**. Third-instar wandering larval eye discs were dissected in phosphate-buffered saline (PBS) and fixed in 4% formaldehyde+1× PBS for 15 min on ice, then in 4% formaldehyde+1× PBS + 0.1% DOC + 0.1% Triton X-100 for 15 min at room temperature (RT). Proteinase K (New England Biolabs) was used at a final concentration of 0.5 µg/mL in PBT (1× PBS+0.1% Tween-20). Digoxigenin (DIG)-labeled RNA probes were denatured at 90 °C for 4 min, using 3–5 µL probe in 25 µL hybridization solution and hybridization was carried out following ThermoFisher FISH Tag RNA Kit suggested hybridization protocol (https://tools.thermofisher.com/content/sfs/manuals/mp32952.pdf). The protocol was modified after Post-Hybridization step 13.7. Mouse anti-DIG (Sigma, #11333062910 (1:250)) was added in PBT and samples were incubated overnight at 4 °C. After overnight incubation, samples were washed 5 times in PBT for 5 min. Anti-mouse secondary Cy3 antibody (Jackson Immunoresearch (1:300)) was added in PBT and incubated for 1 h. Finally, samples were washed three times in PBT for 5 min then mounted in FluorSave (EMD Millipore, #345789) on a glass slide. All steps carried out with gentle rocking on nutator, unless specified. Samples were imaged using Zeiss Confocal microscope.

**DIG-labeled probes**. PCR primers were created for genes, with an average fragment length of 500 bp. Whole third-instar larvae complementary DNA (cDNA)

was amplified by PCR using Q5 High-Fidelity DNA Polymerase (New England Biolabs, #M0491S). DNA purification was performed with QiaGen QIAEX II 20051. A second round of PCR was carried out using 1 ng of cDNA from first PCR, and reverse primers containing the T7 primer; AGGGATCCTAA TACGACTCACTATAGGGCCCGGGGC. T7 RNA polymerase (Sigma, #10881767001) along with DIG RNA labeling mix (Sigma #11277073910) were used for in vitro transcription and incubated for 5–6 h at 37 °C. RNA was then purified using NucleoSpin RNA clean-up (Macherey-Nagel, #740948). Gene primers were as follows: *CG9336*: CCTGAAGTTCGAGGCTGATG, CAACCATT CGTCGTGCATTT, twit: TGAAGCCCATCCATCCAACC, GGCAAGTG TCCCCGAAACTA, *Ldh* (*ImpL3*): ACGGCTCCAACTTTCTGAAG, TTG TTCAACTTCGGTGGGAG, *Ald*: CTCTGAGGATGAGGTCACCA, ATGTTC TCCTTCTTGCCCAGC,

*HIF1A* (*sima*): GAGGGTGGCTTCGAGTTTAG, GGTTTTCCACTCT CCTCTGC.

**Chromatin immunoprecipitation–quantitative polymerase chain reaction**. Fifteen third-instar larvae were collected and homogenized using a tissue grinder with 60 mM KCl, 15 mM NaCl, 4 mM MgCl₂, 15 mM HEPES (pH 7.6), 0.5% Triton X-100, 0.5 mM dithiothreitol (DTT), 10 mM sodium butyrate and protease inhibitor cocktail (Complete, Roche)[37]. Crosslinking was done in 1.8% formaldehyde for 15 min followed by 225 mM Glycine. Lysis buffer is 15 mM HEPES at pH 7.6, 140 mM NaCl, 1 mM EDTA, 0.5 mM EGTA, 0.1% sodium deoxycholate, 1% Triton X-100, 0.5 mM DTT, 0.1% SDS, 0.5% lauroylsarcosine and 10 mM sodium butyrate with protease inhibitor cocktail (Complete, Roche)[37]. Chromatin was sheared using a Branson 450 Sonifier. Antibodies used for immunoprecipitation are rabbit anti-DP (#212, gift from Nick Dyson), mouse anti-Rbf (DX3/DX5, ratio 1:1 gift from Nick Dyson), rabbit anti-E2f1 (#210 gift from Nick Dyson and anti-E2f1 from Gunter Reuter, ratio 1:1), rabbit anti-E2f2 antibodies (#79 gift from Nick Dyson), and rabbit IgG (Sigma) as nonspecific antibody. Samples were pulled down with Protein G Dynabeads (Invitrogen), then washed with lysis buffer four times, with TE (pH 8) twice and eluted. Decrosslinking was done overnight at 65 °C. Samples were treated with RNase A (Sigma) for 1 h at 37 °C followed by proteinase K for 2 h at 50 °C. Then, DNA was purified by phenol–chloroform extraction and overnight ethanol precipitation.

The immunoprecipitated DNA along with the input genomic DNA (collected before precipitation) was quantified by qPCR. SensiFast SYBR No-ROX Mix (Bioline) was used for qPCR. Reactions were run on a LightCycler 480 (Roche). Primer sequences are: *Aldolase* forward: TTTACCGCCCAAACGAAAGC and reverse: GCCAAGCGCTTTAAATTCCC; *sima* (*HIF1A*) forward: AAACG ACCAACGCACATACG and reverse: TTTGGTTCGCGGTGCAATACC. A negative sequence site that does not contain any predicted E2F-binding sites is forward: TGTGTATGCCTTGCTTGCAC and reverse TCTATGCACACGCTCTACTGAG. The protein enrichment was calculated as the percentage of immunoprecipitated DNA relative to input DNA (prior DNA precipitation) for each antibody. Data presented are relative to the negative binding site for each ChIP. Each sample was measured twice. The two-way ANOVA was used to calculate the *p* values.

**Immunofluorescence**. Wandering third-instar larval eye discs were dissected in 1× PBS and fixed in 4% formaldehyde+1× PBS for 15 min, permeabilized in 0.3% PBS-T (1× PBS, 0.3% Triton X-100) two times for 10 min and then incubated with antibodies overnight at 4 °C in 1× PBS+10% normal donkey serum (NDS, Jackson Immunoresearch)+0.1% Triton X-100 blocking serum. The following day, samples were washed in 0.1% PBS-T (1× PBS+0.1% Triton X-100) three times for 5 min. Samples were then incubated with appropriate fluorescently labeled secondary antibodies (Jackson Immunoresearch) for 1 h in 1× PBS+10% NDS+0.1% Triton X-100 followed by 4′,6-diamidino-2-phenylindole (DAPI) for 5 min. Finally, samples were washed five times for 5 min then mounted in FluorSave (EMD Millipore) on glass slides. All steps were carried out at RT and with gentle rocking, unless specified otherwise. Whenever fluorescent images have been compared, they have been obtained with the same acquisition and display settings.

Primary antibodies used were: anti-β-gal: DSHB 40-1a (1:200), anti-cleaved *Drosophila* Dcp-1: Cell Signaling Asp216 (1:500), anti-Dac DSHB mAbdac1-1 (1:100), anti-Elav: DSHB 7E8A10 (1:200), anti-GFP (FITC): Abcam ab6662 (1/1000), anti-Hairy: from T. Orenic (1:4), anti-Hth: from Richard Mann (1:2000), anti-Notch DSHB C458.2H (1:100), anti-Repo: DSHB 8D13 (1:500), anti-Senseless: from H. Bellen (1:100).

**BrdU labeling**. Dissected eye discs were incubated with Schneiders media (Fisher, 21720024) containing BrdU at a final concentration of 0.2 mg/mL for 2 h at RT. The tissues were fixed in 1.5% formaldehyde+0.01% Tween-20+1× PBS overnight at 4 °C. Tissues were washed twice with 1× PBS twice for 30 min. The eye discs were digested using 5 μL DNAse (Promega RQ1, #M6101) in 1× PBS at a total volume of 100 μL and incubated for 40 min at 37 °C, then washed twice for 5 min with 1× PBS+0.3% Triton. Tissues were blocked for an hour (see Immuno-fluorescence) and then the BrdU antibody was added (1:50 (BD Biosciences,

347580)). The Immunofluorescence protocol was followed from this step, see above.

**Cleaved Dcp-1 quantification**. Confocal images were inverted and cropped in Adobe Photoshop to remove antennal discs, reducing background staining. The following filters were applied: Brightness +125, Contrast +50, levels: 0, 1.0, 220. Dcp-1-positive cells were then counted using ImageJ software using conservative parameters in order to minimize detection of background; Threshold, Black-Background is false, Convert to Mask, Fill Holes, Watershed, Analyze Particles (size = 50–Infinity). The same parameters were used for all eye discs in each genotype. Quantification was done using two RNAi lines for *Ald* (BL#26301 and BL#65883) and counting 18 and 42 eye discs respectively, and *HIF1A/sima* (BL#33894 (48 eye discs), BL#33895 (11 eye discs)) and the datasets combined. A total number of 62 *Rbf*[120a] and 54 *Ldh* knockdown eyes were used. Pooled data were analyzed in R using one-way ANOVA followed by Tukey's HSD post-hoc test of significance. A box plot was created, individual eyes plotted as gray circles and outliers as white circles.

**Live pH staining**. Third-instar larval eye discs were dissected and washed twice with 1× PBS (pH 7.4) Then, it was incubated with 1 μL of pHrodo Red AM Intracellular dye (Life Technologies cat. no. P35372) Indicator probe+98 μL 1× PBS (pH 7.4)+1 μL of the buffer included in pHrodo for 25 min. The eye discs were mounted in FluorSave (EMD Millipore #345789) on a glass slide then imaged using Zeiss Confocal microscope.

**Tissue dissociation for Drop-seq**. Eye discs were dissected within 1 h and then transferred to a microcentrifuge tube where they were dissociated in a final concentration of 2.5 mg/mL Collagenase (Sigma #C9891) and 1× trypsin (Sigma #59418C) in Rinaldini solution. The microcentrifuge tube was horizontally positioned on a 225 rpm shaker for 20 min at RT. This dissociation protocol resulted in healthy single cells and less than 5% clumps.

**Drop-seq**. We followed the Drop-seq protocol[7] Online-Drop-seq-Protocol-v.-3.1-Dec-2015.pdf at http://mccarrolllab.com/dropseq/ while having four modifications. the final concentration of Sarkosyl in the lysis buffer was 0.4%. The cycles in the PCR step post exonuclease are 4 cycles, and then 12. The cDNA Post PCR was purified twice with 0.6× AMPure beads. The tagmented DNA for sequencing was purified twice: first using 0.6× AMPure beads and the second time using 1× AMPure beads.

**scRNA-seq data analysis**. Illumina paired end raw sequences (FastQ file) were processed for read alignment and gene expression quantification. Drop-seq single-cell data were analyzed using the data analysis protocol described in Drop-seq cook-book (version 1.2 Jan 2016)[7] (http://mccarrolllab.com/dropseq/) and used the Drop-seq_tools-1.13. We used STAR aligner to align the reads against *Drosophila melanogaster* genome version BDGP6 (Ensembl) and corresponding gene model was extracted from Ensembl version 90. Quality of reads and mapping were checked using the program FastQC (https://www.bioinformatics.babraham.ac.uk/projects/fastqc/).

Digital Gene Expression (DGE) matrix data obtained from an aligned library is done using the Drop-seq program DigitalExpression (integrated in Drop-seq_tools-1.13). Number of cells that were extracted from aligned BAM file is based on knee plot which extracts the number of reads per cell, then plot the cumulative distribution of reads and select the knee of the distribution.

Drop-seq alignment was performed as mentioned in Drop-seqAlignmentCookbookv1.2Jan2016 at http://mccarrolllab.com/dropseq/. Alignment was performed using the *Drosophila melanogaster* BDGP6 genome while excluding mitochondria encoded genes. Quality checks were performed then matrices were created for downstream bioinformatic analyses.

**Cell clustering and discovery of cell types**. The following computational figures: dotplots, feature plots, tSNE, heatmaps and gene/gene plots were generated using Seurat[8].

Following alignment, the extracted gene matrices were subjected to unsupervised identify probable cell types using R package Seurat (R version 3.3.2, Seurat version v2.2.1). In Seurat analysis we performed initial quality control analysis and low-quality cells were filtered out using 200 and 3000 gene/cell as a low and top cutoff, respectively, min.cell = 3 and total.expr = 1e4. Statistically significant principal components were determined using JackStraw procedure. The first 20 principle components were selected to run non-linear dimensional reduction (tSNE) while a resolution of 1.6 was applied in both WT (11,416 cells) and WT+*Rbf* (5591+5203) analyses. The WT analysis was performed only on the cell barcodes as listed in WT.xlsx (Supplementary Data 8) whereas the WT+Rbf analysis was performed on WT_Rbf.txt barcodes listed in text file WT_Rbf.xlsx (Supplementary Data 9). These files exclude cells having a predominant ribosomal gene expression signature coming from each sample. Following a resolution of 1.6,

the interommatidial cells (INT) split into 3 clusters in the WT analysis. Two of these have the same top markers indicating an artifact from Seurat analysis and, as a result, were both labeled as INT. The third population has a cell-cycle signature with top markers such as *Claspin*, *stg* and *PCNA*. This population was therefore labeled as SMW. Populations having heat shock proteins as top markers and any remaining ribosomal proteins were grouped and labeled as other in dark gray.

**Gene set enrichment analysis.** Gene set enrichment analysis (GSEA) is a computational method that determines whether an a priori defined set of genes shows statistically significant, concordant differences between two biological states. We applied this statistical analysis to find which Gene Ontology (GOBP) shows statistically significant, concordant differences in each cell cluster.

Using average expression of genes in each cell cluster, we selected UND, EPR and LPR populations for GSEA analysis. We calculated the log2 fold change $(log2FC) = log2(B) - log2(A)$; where $B$ is the average expression value of a specific gene in the cluster of interest and $A$ is the median expression value of that gene in the other clusters. The log2FC values were then ranked and ranked genes matched to categories of GOBP (http://www.go2msig.org/cgi-bin/prebuilt.cgi?taxid=7227). The normalized enrichment score for each process was calculated based on the Broad Institute's GSEA directions after 10,000 permutations to obtain accurate false discovery rate (FDR) values (http://software.broadinstitute.org/gsea/index.jsp). We used gitools v1.8.4 to create the heatmap while using FDR < 0.05 as the cutoff for significantly enriched, upregulated biological processes. If the processes are down-regulated, they were labeled as FDR = 1.

**Monocle 2 trajectory.** Single-cell pseudotime trajectory was constructed using Monocle 2[29] (monocle version 2.8.0, with R version 3.5.0) using UND, PPN, MF, INT, EPR and LPR cells from the WT scRNA-seq data. The following parameters were used to generate the plot: top 1500 genes were selected, Rho = 20 and Delta = 20, *q*val < 0.01 and DDRTree dimension reduction method.

## Data availability

The authors declare that all data supporting the findings of this study are available within the article and its supplementary information files or from the corresponding author upon reasonable request. All raw and processed data have been deposited in the Gene Expression Omnibus (GEO) database under accession code GSE115476. Raw and processed data were also deposited in the European Bioinformatics Institute ArrayExpress with the accession numbers E-MTAB-7194 and E-MTAB-7195. Interactive tSNEs were submitted to the Broad Institute single cell portal (https://portals.broadinstitute.org/single_cell) for the wild-type and wild-type/Rbf analyses. The source data underlying Figs. 1b and 3b and Supplementary Tables 2 and 3 are provided as a Source Data file.

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

## Acknowledgements

We are grateful to Oni Basu, Nick Dyson, Nissim Hay and Kristin White for discussion, to Carlotta Rubio-Perez for help with GOBP analysis, to Stein Aerts for sharing unpublished data and to B. Paterson and the Developmental Studies Hybridoma Bank for antibodies. This work was supported by the National Institutes of Health grants GM93827 and GM110018 (to M.V.F.).

## Author contributions

M.M.A. and M.V.F. conceived the project, designed the experiments, analyzed data and wrote the manuscript. M.M.A. performed most of the experiments and Seurat analysis with important support by M.C. who performed immunostainings and Dcp-1 quantification. A.B.M.M.K.I. established, supervised and performed the bioinformatics analysis. M.P.Z. performed ChIP-qPCR assays. All authors reviewed and edited the manuscript.

## Additional information

**Competing interests:** The authors declare no competing interests.

