## [Peer Review File · Nature Communications]

Reviewers' Comments:

Reviewer #1:

Remarks to the Author:

The eye-antennal imaginal disc of the fruit fly can be used as powerful genetic model for tissue development and cell fate determination. In the current manuscript, the authors use the eye disc to investigate mechanisms of how the retina blastoma tumor suppressor (the fly homolog is Rbf) contributes to tissue overgrowth. To address this a scRNAseq approach is used, identifying a molecular link between Rbf to apoptosis and cellular metabolites.

The manuscript is based on two complementary aspects: first, the single cell sequencing approach to assess the eye-disc model, and second, the identification of genetic mechanisms that are involved in Rbf mediated defects. While, conceptually this is a very nice proof of concept to use scRNAseq to address a central biological question it also is partly a weakness of the manuscript. The scRNAseq analysis does not account for the impact it might have in the field. While I feel this is an important contribution I would highly encourage the authors to go beyond their current analysis to provide a cornerstone for future scRNAseq analyses of the eye-antennal imaginal disc.

Minor comments.

- Currently the authors use Seurat for dimensionality reduction. While this is indeed a powerful tool alternative tools give complementary information. In particular SCENIC and Seurat are currently often used side-by-side.

- A main point of the wildtype scRNAseq analysis is the identification of a switch in the transcriptional program from early to late photoreceptor precursors. This is an interesting point, which is not further explored, except in the context of the Rbf mutation. It would be worth to describe this temporally in more detail and to provide clearer evidence for the time point (after MF) where this occurs. There are numerous markers in literature that can be used to depict this precisely. Which leads me to another minor point: this use of markers would be beneficial for the analysis of all the newly identified genes. As mentioned above the wildtype description here is pioneering and will be as reference for future investigations in the field, thus should provide a corner stone.

- Not all cell types are depicted in figure 1e, it would be interesting to either clarify why or (even better) to display these clusters as well.

- What is "Ace", which used as marker for LPR? Expression co-staining with PR or furrow markers would help to identify where it is expressed.

- The section of the Rbf mutant analysis appears quite short, while the authors apparently refer to much more information that is clearly shown. Please extend.

Reviewer #2:

Remarks to the Author:

Ariss et al. use single-cell transcriptomics to construct a molecular atlas of the drosophila eye disc cells at 1x cellular coverage. They use extensive available ground truth to validate this atlas and identify novel markers. They report a "transcriptional switch" in differentiating photoreceptors at the time of axonogenesis. Next, they use the healthy atlas as a scaffold to map the cell landscape of Rbf mutant eye disc, and identify a mutant-specific cell population that exhibits a molecular signature suggesting intracellular acidification due to increased glycolytic activity. They test this observation with genetic experiments, and show that these cells are apoptotic.

Overall I feel very positive about this story and the attempt to link scRNA-seq to functional phenotype in a mutant. To that extent, I am reasonably enthusiastic about this paper. However, I do have a number of issues that I would like the authors to address in order to solidify some of the analysis as well as the biological message of their story before it's ready for publication. My comments follow,

1. It's not clear at what developmental stage the reported healthy atlas corresponds to. I am assuming it's one of the larval stages but maybe I missed an explicit mentioning in the text.

2. It would be good to have a figure supporting the claim that "cells from each of 11 biological replicates were evenly distributed among the cell populations"

3. While I agree with the result in Fig. 1d, I disagree with its presentation. Coexpression of two individual markers (or the lack of it) in single-cell RNA-seq data is confounded with the efficiency of detection. Thus, it's possible that even two truly coexpressed markers would show low co-expression if they are expressed at low levels. Thus, it's not just the absolute % of cells that coexpress a given marker, but the ratio of the observed vs. expected proportion of co-expressing cells, where the "expected" proportion can be calculated by assuming independent expression.

For example, if 20% of photoreceptors express Sens and 40% of photoreceptors express BarH2, then the expected proportion of photoreceptors that co-express these two (based on independent expression) is 8%. The observed expression should be significantly less than this value to suggest mutual exclusion. It would be good if the authors can include such analysis in their arguments.

4. "Cells from SMW exhibited high expression of cell cycle genes such as Claspin, PCNA,, whilst INT cells did not. We concluded that INT are quiescent interommatidial cells, whereas SMW consists of cells from the second mitotic wave" - these seems to be a completely unjustified statement. I did not see any evidence of cell cycle gene expression. Also, SMW appears to be so proximal to INT in the tSNE plot, I am wondering if it really represents a discrete subset. Also, why does Fig. 1e (or an analogous supplementary figure) not include all clusters in Fig. 1c, if they indeed represent transcriptionally distinct subsets?

5. "PG specific gene CG3168 was also exclusively detected in this population" - then why is it not highlighted in Fig. 1? In the same vein, the authors could include a supplementary figure showing the expression of the markers NK7.1 and CG9336. More generally, I would request the authors to include supplementary figures to illustrate all cell type specific marker expression claimed from the data.

6. I do not understand the statement in the legend of Fig. 2a - "The percentage represents the cells co-expressing both genes" nor the statement in the main text "notably over 80% of cells showed coexpression of each aforementioned marker with elav". What's been shown in the panels in Fig. 2a is that 76% of sens+ cells in EPR express elav or 81% of ro+ cells express elav in LPR. The inverse question, which is not clear, is what % of all cells are +ve for elav with one or more of these markers. Is that >80%? Then it would be consistent with the authors claims.

Furthermore, elav is a highly expressed gene so it's not very surprising that most photoreceptors express elav. However, what is surprising that both EPR and LPR contain cells that express all R-cell specific markers, suggesting that the major transcriptional variation in the photoreceptor population is early vs late. Which makes this reviewer wonder whether within EPR or LPR, is there a segregation between sens+ or ro+ or B-H2+ cells? for example, are there additional transcriptional differences between ro+ cells and B-H2+ cells within EPR? Have the authors tried to analyze the LPR population alone to see if they are able to subcluster different photoreceptor types?

7. I am not a drosophila biologist by any means (in case that wasn't clear until now) but I am failing to see the correspondence in the stains between Fig. 3d and 4a. The authors claim that Cluster 13, localized by the expression, of Ldh (which is pretty specific based on Fig. 3c) is the same as the apoptotic Dcp-1+ cells in Fig. 4a. I see the two patterns as very different. Are these the same cells?

It's entirely possible that the Ldh-mediated apoptosis might be a cell non-autonomous effect. Have the authors considered this possibility? Thus cluster 13 might represent the cell type responsible for the apoptosis of other cells.

9. Related to 7, did the authors check the relative proportions of various cell types in the mutant vs wild-type data?

10. Finally, in the author's own words, this work represents an important resource for the drosophila eye disc community. It would be great if they can also make this resource available to others by sharing their data in an accessible format. Have they considered uploading their data to the single-cell portal or on an alternative website, where interested researchers can query gene expression patterns?

Reviewer #3:

Remarks to the Author:

This work describes the use of single cell RNA-sequencing (scRNA-seq) to profile cell types of the *Drosophila* eye imaginal disc (ED). Single cell transcriptional profiles were clustered using a t-stochastic neighbor embedding (tSNE) analysis method, the clusters were assigned to eye-disc cell types based on known expressed markers, and then the scRNA-seq data was then used to demonstrate that photoreceptors undergo a transcriptional switch, from early photoreceptors to late photoreceptors in WT discs. This is expected, but this is the first detail examination of the process using this new method. Next, the authors performed another scRNA-seq experiment, this time using Rbf mutant cells, whose transcriptional profile was compared with WT transcriptional profile. They discovered a cluster of cells that was specific to Rbf mutant discs. Cells of this cluster, termed "cluster 13," displayed enriched expression of glycolytic regulators; Aldolase, HIF1a and lactate dehydrogenase (ImpL3). Consequently, these cells displayed lower pH, due to increased glycolytic activity. The region harbouring this cell cluster also displayed increased apoptosis, suggesting that Rbf loss may promote apoptosis through altered metabolism. Although Rb loss has long been known to promote apoptosis, the underlying mechanism remains controversial, and this metabolic explanation is somewhat novel.

We appreciate that the large dataset presented here is of great value and may pave the way for new, interesting observations related to eye development and Rbf function. The main bulk of this work revolves around developing scRNA-seq to analyze tissues, eyedisks, composed of multiple cell types. The technique is very elegant in that it allows us to dissect the transcriptome of specific cell types within a complex tissue. Two, unrelated, applications of this dataset are shown here. First, the authors demonstrated how this dataset/technique may be used to uncover transcriptional transitions/switching between differentiating cell populations (early and late photoreceptors). This observation by itself is not very interesting, as it is no longer novel either for the ED or for scRNAseq, but nevertheless the demonstration serves to illuminate some applications and limitations of this technique/dataset. Second, the authors compared scRNA-seq from WT cells with Rbf mutant cells. The tSNE clustering revealed a Rbf mutant specific cluster, that was enriched in glycolytic genes. The observation that Rbf loss promotes apoptosis dependent on glycolysis is novel and interesting. It is also well validated by genetic and chemical tests. Given the high quality of the analysis we would support publication in Nature Communications.

Minor comments and questions:

1. Figure 1a,c, Figure 2f and figure 3a: It would be helpful if the cell types were colored consistently between all figures.
2. Figure 1c and figure 3a: I am not familiar how tSNE works, but why is the map in figure 3a not rotated similar to the map in figure 1c: (see picture). Is it possible to align them similarly? It also seems like the VPE clusters and OCx clusters have switched positions between these two plots (OCx aligns closer to DPE in Fig1c, consistent with note made in supplementary table 1). Is this by mistake or is it by chance due to similarities between these clusters?
3. Left: Figure 3a, Right Figure 1c.
4. Figure 2f: What are component 1 and 2? Please define these if possible.
5. Figure 3b. I think an outline of approximate cluster borders would be useful.
6. I noticed that there were a lot of other interesting changes seen in the Rbf mutants, such as decreased amounts of photoreceptor cells. Are there also increased amounts of pedipodial epithelium cells? Please comment on the other mutant-specific changes observed.
7. Figure 4B. Ald, HIF1a and Ldh RNAi seem to reduce apoptosis, but there is still a lot of apoptosis going on. Could this be due to efficiency of the RNAi's, or is apoptosis affected by reduced metabolism as a consequence of reduced glycolytic activity? Can apoptosis be induced by overexpression of Ldh, HIF1a or Ldh in non-mutant cells?
8. Figure 4C. E2F1 seems to interact weakly with Ald and HIF1a compared to Rbf, Dp and E2F2. What larval tissues have been used? Cycling or quiescent tissue? Are the differences measured significantly different (p values and which statistical test)? Do error bars represent Standard deviation or SEM?
9. Figure 4D and discussion. They claim lower pH results in apoptosis. They should artificially increase the pH (through Ldh over-expression?), and look for increased apoptosis in the cells with low EGFR activity.

In order to prevent any confusion, be advised the Figures 2, 3 and 4 mentioned in the reviews are now Figures 3, 4 and 5, accordingly. Reviewers' comments and questions are italicized.

Reviewer #1

“The manuscript is based on two complementary aspects: first, the single cell sequencing approach to assess the eye-disc model, and second, the identification of genetic mechanisms that are involved in Rbf mediated defects. While, conceptually this is a very nice proof of concept to use scRNAseq to address a central biological question it also is partly a weakness of the manuscript. The scRNAseq analysis does not account for the impact it might have in the field. While I feel this is an important contribution I would highly encourage the authors to go beyond their current analysis to provide a cornerstone for future scRNAseq analyses of the eye-antennal imaginal disc.”

Response: We are very pleased that the reviewer appreciated the potential impact of our manuscript and thank the reviewer for suggestions on how to improve it. As the reviewer suggested, we expanded on the section describing the wild type eye disc that we hope will contribute to the effort in the Drosophila community to build a cell atlas of the wild type larval eye disc. A new Figure 2 now shows novel markers for different cell populations that we discovered in this work. More details are provided in our response to point 2 below.

Minor comments:

Point 1. *“Currently the authors use Seurat for dimensionality reduction. While this is indeed a powerful tool alternative tools give complementary information. In particular SCENIC and Seurat are currently often used side-by-side.”*

Response: [redacted]

Point 2. *“A main point of the wildtype scRNAseq analysis is the identification of a switch in the transcriptional program from early to late photoreceptor precursors. This is an interesting point, which is not further explored, except in the context of the Rbf mutation. It would be worth to describe this temporally in more detail and to provide clearer evidence for the time point (after MF) where this occurs. There are numerous markers in literature that can be used to depict this precisely”*

Response: First, to further characterize this transcriptional switch and distinctions between LPR and EPR cells, we performed a Seurat analysis separately on EPR and LPR populations. Interestingly, we find no evidence of clustering photoreceptors by R types. This can be seen by the random expression pattern of *sens*, *ro*, and *B-H2* in the feature plots. We added these results in **Supplementary Figure 1b**.

Second, to address the reviewer's question at what point in the eye development the transcriptional switch occurs, we co-immunostained *Ace*-GFP (a marker for LPR) with Sens antibodies to show that LPR is detected after column 4 (**Figure 3d**) described on page 11.

"Which leads me to another minor point: this use of markers would be beneficial for the analysis of all the newly identified genes. As mentioned above the wildtype description here is pioneering and will be as reference for future investigations in the field, thus should provide a corner stone." We appreciate that the reviewer values our work and we included the following additional data to address his question. This new information is listed below:

Response: First, we generated a dotplot in **Figure 2a** showing top markers in each population. This figure contains 34 known markers, 9 markers previously studied that were not specified in which cell type/domain they belonged to, and finally 26 novel, differentially expressed markers that strongly show a biased expression pattern towards a specific population(s). In some instances, these markers show a stronger and more biased expression pattern than classical markers (e.g. *s/s* vs. *g/l*). We validated the expression of several of these genes by immunofluorescence (**Fig. 2**) (see the next point). These 26 genes could also be playing a role in the specific cell type they belong to and affect eye development. Since they were previously not shown to be expressed in the eye disc, they are potential candidates for future investigation. These new data are described on pages 8-10.

Second, we added a BrdU labeling in **Figure 1b** as part of the description for the eye disc. This figure points at the SMW, which are interommatidial (INT) cells undergoing a round of cell division, as well as the dividing perineurial glial (PG) cells. We also added stainings for *s/s*, *cpo*, and *pigs* reporters. Based on the scRNA-seq results, *s/s* (also known as *kettin*) is a posterior marker. We support that finding in the immunostaining in **Figure 2b**. This is surprising since *s/s* is a muscle gene, and our results show that the gene is also expressed in the eye disc. We also found that *cpo*, which was previously thought to be expressed posterior to the morphogenetic furrow (Harvie et. al (1998)), is only seen in photoreceptors on the eye disc proper as well as subperineurial and wrapping glia in the basal compartment of the disc. This is supported by the dotplot and the stainings in **Figure 2** and is described on pages 8-10. We also discovered an interesting expression pattern of the gene *pigs*. The bioinformatic and staining result shows that the gene is expressed in perineurial glia and hemocytes (**Figure 2** and described on pages 8-10).

Point 3. *"Not all cell types are depicted in figure 1e, it would be interesting to either clarify why or (even better) to display these clusters as well."*

Response: As requested, a new **Figure 1e** shows a full heatmap to visualize each population along with the top markers. These data are described on page 7.

Point 4. *"What is "Ace", which used as marker for LPR? Expression co-staining with PR or furrow markers would help to identify where it is expressed."*

Response: We added a description of *Ace* in the text on page 11. We also co-stained *Ace*-GFP with Sens to show that it is expressed in photoreceptors after column 4 as well as in axonal projections. We added an arrow to point at the axons (**Fig. 3d**).

Point 5. “*The section of the Rbf mutant analysis appears quite short, while the authors apparently refer to much more information that is clearly shown. Please extend.*”

Response: As per the reviewer’s request we extended the description of *Rbf* mutant analysis on page 12-13). We also included a new piece of data that shows that overexpression of *hid* is insufficient to trigger intracellular acidification (**Supplementary Fig. 3**) in contrast to what we discovered in *Rbf* mutant. This is an important conceptual point since *hid* is a key target of Rb pathway in induction of apoptosis. Thus, intracellular acidification of *Rbf* mutant cells is not merely the result of *hid* induction. We discussed slight changes in number of *Rbf* mutant cells that contribute to different cell populations (page 12). We expanded on discussion whether acidification causes apoptosis in cell autonomous manner (page 17).

Reviewer #2

“Ariss et al. use single-cell transcriptomics to construct a molecular atlas of the drosophila eye disc cells at 1x cellular coverage. They use extensive available ground truth to validate this atlas and identify novel markers. They report a "transcriptional switch" in differentiating photoreceptors at the time of axonogenesis. Next, they use the healthy atlas as a scaffold to map the cell landscape of Rbf mutant eye disc, and identify a mutant-specific cell population that exhibits a molecular signature suggesting intracellular acidification due to increased glycolytic activity. They test this observation with genetic experiments, and show that these cells are apoptotic.

Overall I feel very positive about this story and the attempt to link scRNA-seq to functional phenotype in a mutant. To that extent, i am reasonably enthusiastic about this paper. However, I do have a number of issues that I would like the authors to address in order to solidify some of the analysis as well as the biological message of their story before it's ready for publication. My comments follow,”

We thank the reviewer for his/her comments and we are pleased that the reviewer feels so positive about our manuscript. Specific points are addressed below:

Point 1. *“It's not clear at what developmental stage the reported healthy atlas corresponds to. I am assuming it's one of the larval stages but maybe I missed an explicit mentioning in the text.”*

Response: We used the third instar larval eye disc in this work. The developmental stage is mentioned on page 5, at the beginning of the Results section.

Point 2. *“It would be good to have a figure supporting the claim that "cells from each of 11 biological replicates were evenly distributed among the cell populations. “*

Response: We included **Supplementary Figure 1a** showing the cells from each biological replicate being distributed throughout the tSNE plot.

Point 3. *“While I agree with the result in Fig. 1d, I disagree with its presentation. Coexpression of two individual markers (or the lack of it) in single-cell RNA-seq data is confounded with the efficiency of detection. Thus, it's possible that even two truly coexpressed markers would show low co-expression if they they are expressed at low levels. Thus, it's not just the absolute % of cells that coexpress a given marker, but the ratio of the observed vs. expected proportion of co-expressing cells, where the "expected" proportion can be calculated by assuming independent expression.*

For example, if 20% of photoreceptors express Sens and 40% of photoreceptors express BarH2, then the expected proportion of photoreceptors that co-express these two (based on independent expression) is 8%. The observed expression should be significantly less than this value to suggest mutual exclusion. It would be good if the authors can include such analysis in their arguments. “

Response: We thank the reviewer for his/her comment about the presentation. As the reviewer suggested, we calculated the expected co-expression frequency within *sens+* and *ro+* cells of based on independent expression. The observed co-expressing percentage of *sens* and *ro*

positive cells within all *sens+* and *ro+* cells is 1.7%. In this example, $p(\text{sens}) = 27.17\%$ and $p(\text{ro}) = 72.83\%$. Therefore, the expected % co-expression based on independent expression is 19.79%. The observed value is lower than expected ($1.7\% < 19.79\%$).

The same was performed with *sens+* and *B-H2+* cells, where the expected co-expression was 16.8% based on independent expression. The observed value in these cells was 3.6%, which is lower than the expected ($3.6\% < 16.8\%$)

We performed chi-squared test using these two observed and expected values, meaning the degree of freedom is 1. The results showed that the observed values do not fit the expected model and are statistically significantly lower: p value of 2.3×10^{-7} . We added in the manuscript that the expected % was calculated and was significantly lower on page 6-7.

Point 4. *"Cells from SMW exhibited high expression of cell cycle genes such as Claspin, PCNA, ..., whilst INT cells did not. We concluded that INT are quiescent interommatidial cells, whereas SMW consists of cells from the second mitotic wave" - these seems to be a completely unjustified statement. I did not see any evidence of cell cycle gene expression. Also, SMW appears to be so proximal to INT in the tSNE plot, I am wondering if it really represents a discrete subset. Also, why does Fig. 1e (or an analogous supplementary figure) not include all clusters in Fig. 1c, if they indeed represent transcriptionally distinct subsets?"*

Response: We apologize for the confusion caused. In the *Drosophila* eye disc, interommatidial cells that undergo a synchronous round of cell division posterior to the morphogenetic furrow are called Second mitotic wave. To explicitly state this, we added in the manuscript that the INT and SMW populations are both "interommatidial cells" (page 5 and 7-8). This also explains the close proximity in the tSNE between both populations as they are essentially different by the expression of cell cycle genes. We have generated a dotplot to depict the difference in expression of cell cycle genes such as *Claspin*, *PCNA* and *Mcm7* between INT and SMW (**Fig. 2b**). We also added a BrdU labeled eye disc in **Figure 1b** that shows second mitotic wave where the interommatidial cells that are undergoing a final round of cell division. Additionally, the reviewer asked for the full heatmap, so we replaced **Figure 1e** with a heatmap showing the top markers for all clusters.

Point 5. *"PG specific gene CG3168 was also exclusively detected in this population" - then why is it not highlighted in Fig. 1? In the same vein, the authors could include a supplementary figure showing the expression of the markers NK7.1 and CG9336. More generally, I would request the authors to include supplementary figures to illustrate all cell type specific marker expression claimed from the data. "*

Response: As per the reviewer's request, we added NK7.1 and CG9336 in **Figure 2a**; NK7.1 is most highly expressed in WG+SPG, whereas CG9336 is expressed exclusively in this cluster. We also show expression of CG3168 in the **Figure 2a**. We generated a dotplot of the top known and unknown markers for all populations (**Fig. 2a**) and this figure is described in the legend on page 35.

Point 6. *"I do not understand the statement in the legend of Fig. 2a - "The percentage represents the cells co-expressing both genes" nor the statement in the main text "notably over 80% of cells showed coexpression of each aforementioned marker with elav". What's been shown in the panels in Fig. 2a is that 76% of sens+ cells in EPR express elav or 81% of ro+*

cells express *elav* in LPR. The inverse question, which is not clear, is what % of all cells are +ve for *elav* with one or more of these markers. Is that >80% ? Then it would be consistent with the authors claims.

Furthermore, *elav* is a highly expressed gene so it's not very surprising that most photoreceptors express *elav*. However, what is surprising that both EPR and LPR contain cells that express all R-cell specific markers, suggesting that the major transcriptional variation in the photoreceptor population is early vs late. Which makes this reviewer wonder whether within EPR or LPR, is there a segregation between *sens+* or *ro+* or *B-H2+* cells? for example, are there additional transcriptional differences between *ro+* cells and *B-H2+* cells within EPR? Have the authors tried to analyze the LPR population alone to see if they are able to subcluster different photoreceptor types? "

Response: We thank the reviewer for these comments. To address them, we removed the % in the gene/genes plots for in **Figure 3a** and instead describe that both EPR and LPR share R-cell type markers on page 10. As the reviewer suggested, we performed a supervised Seurat analysis by selecting EPR and LPR cells in order to determine whether there is any segregation between *sens+*, *ro+*, or *B-H2+* within EPR. The feature plots in **Supplementary Figure 1b** shows that these genes are randomly expressed throughout the plots without any specific segregation pattern, therefore photoreceptors do not cluster by R type.

Point 7. "I am not a drosophila biologist by any means (in case that wasn't clear until now) but I am failing to see the correspondence in the stains between Fig. 3d and 4a. The authors claim that Cluster 13, localized by the expression, of *Ldh* (which is pretty specific based on Fig. 3c) is the same as the apoptotic *Dcp-1+* cells in Fig. 4a. I see the two patterns as very different. Are these the same cells?

It's entirely possible that the Ldh-mediated apoptosis might be a cell non-autonomous effect. Have the authors considered this possibility? Thus cluster 13 might represent the cell type responsible for the apoptosis of other cells. "

Response: The reviewer raises an important point whether the cells that express glycolytic genes and therefore have increased intracellular acidification are exactly the same cells that are *Dcp-1* positive (apoptotic). Since the pHrodo is a live dye it precludes performing co-staining with *Dcp-1* antibody that is done on fixed tissues to directly address this point. Although we cannot completely exclude the possibility that non-glycolytic cells undergo apoptosis, our data are consistent with the interpretation that intracellular acidification causes apoptosis in cell autonomous manner. First, the cells with low pH (**Fig. 4e**) and *Dcp-1* staining (**Fig. 5a**) are localized to the same domain, anterior to the morphogenetic furrow (MF). To make this point clear, we added an MF arrowhead in all figures including **Figure 1a**, the eye disc map to help guide and locate where the MF. All eye disc images in the manuscript are positioned so anterior is to the left of the MF. Second, cluster 13 cells (which expresses *Ldh*, *Ald*, *HIF1a*) also expresses *hid* - a pro-apoptotic gene (**Fig. 4c**). *hid* has been previously shown to be critical for apoptosis in *Rbf* mutants (Moon et al 2006). If the process was cell non-autonomous, we would see a different pattern where *hid* would be the marker of a population other than cluster 13. We have elaborated on this point in Discussion on page 17.

Additionally, if the effect was non-autonomous, we assume that lactate would be transported outside the cell and decrease pH in other cells. So far, we found no evidence of elevated expression pattern of lactic acid and monocarboxylate transporters (MCT) within the WT and

Rbf^{20a} mutants; CG11665, CG14196, CG8028, CG8034, CG8051, *out*, CG8389, *Sln*, CG13907, CG8486, *Mct1*. We observed a consistently low level of expression of these genes across all populations. We understand that the MCT transport activity does not rely solely on the RNA levels of the transporters, nonetheless this supports the argument that the effect is most likely autonomous as there is no change in MCT gene expression.

Point 9. *“Related to 7, did the authors check the relative proportions of various cell types in the mutant vs wild-type data?”*

Response: We attempted this analysis but it proved to be uninformative. This primarily because the morphogenetic furrow is continuously moving during the eye development and separates undifferentiated cells in the anterior from differentiating photoreceptors. This affects the relative proportions of cells in different clusters. For example, the number of photoreceptors is defined by the current position of the morphogenetic furrow. Since *Rbf* mutants exhibit a slight development delay, the MF slightly lags behind and therefore the number of photoreceptors appear to be lower than in the wild type. We mentioned this on page 12. Please, also see our response in Reviewer 3’s point 6.

Point 10. *“Finally, in the author’s own words, this work represents an important resource for the drosophila eye disc community. It would be great if they can also make this resource available to others by sharing their data in an accessible format. Have they considered uploading their data to the single-cell portal or on an alternative website, where interested researchers can query gene expression patterns?”*

Response: We have submitted the data to GEO with the accession number GSE115476 and, as per the reviewer’s request, we are also currently in the process of submitting our data set to the EBI Single Cell Expression Atlas. We are in contact with Anja Fullgrabe at EBI to make the data accessible.

Reviewer #3

“This work describes the use of single cell RNA-sequencing (scRNA-seq) to profile cell types of the Drosophila eye imaginal disc (ED). Single cell transcriptional profiles were clustered using a t-stochastic neighbor embedding (tSNE) analysis method, the clusters were assigned to eye-disc cell types based on known expressed markers, and then the scRNA-seq data was then used to demonstrate that photoreceptors undergo a transcriptional switch, from early photoreceptors to late photoreceptors in WT discs. This is expected, but this is the first detail examination of the process using this new method. Next, the authors performed another scRNA-seq experiment, this time using Rbf mutant cells, whose transcriptional profile was compared with WT transcriptional profile. They discovered a cluster of cells that was specific to Rbf mutant discs. Cells of this cluster, termed "cluster 13," displayed enriched expression of glycolytic regulators; Aldolase, HIF1a and lactate dehydrogenase (ImpL3). Consequently, these cells displayed lower pH, due to increased glycolytic activity. The region harbouring this cell cluster also displayed increased apoptosis, suggesting that Rbf loss may promote apoptosis through altered metabolism. Although Rb loss has long been known to promote apoptosis, the underlying mechanism remains controversial, and this metabolic explanation is somewhat novel. “

We appreciate that the large dataset presented here is of great value and may pave the way for new, interesting observations related to eye development and Rbf function. The main bulk of this work revolves around developing scRNA-seq to analyze tissues, eyedisks, composed of multiple cell types. The technique is very elegant in that it allows us to dissect the transcriptome of specific cell types within a complex tissue. Two, unrelated, applications of this dataset are shown here. First, the authors demonstrated how this dataset/technique may be used to uncover transcriptional transitions/switching between differentiating cell populations (early and late photoreceptors). This observation by itself is not very interesting, as it is no longer novel either for the ED or for scRNAseq, but nevertheless the demonstration serves to illuminate some applications and limitations of this technique/dataset. Second, the authors compared scRNA-seq from WT cells with Rbf mutant cells. The tSNE clustering revealed a Rbf mutant specific cluster, that was enriched in glycolytic genes. The observation that Rbf loss promotes apoptosis dependent on glycolysis is novel and interesting. It is also well validated by genetic and chemical tests. Given the high quality of the analysis we would support publication in Nature Communications.

We thank the reviewer for his/her enthusiasm about our work and especially that the reviewer appreciated the value of our findings for Rb field.

Minor comments and questions raised by the reviewer:

Point 1. *“Figure 1a,c, Figure 2f and figure 3a: It would be helpful if the cell types were colored consistently between all figures. “*

Response: We have modified the colors of cell types in the eye map (**Fig. 1a**) and trajectory (**Fig. 3f**) to match the colors from the tSNEs (**Fig. 1c** and **4a**). Now all the population colors are consistent throughout the manuscript.

Point 2. *“Figure 1c and figure 3a: I am not familiar how tSNE works, but why is the map in figure 3a not rotated similar to the map in figure 1c: (see picture). Is it possible to align them similarly?”*

It also seems like the VPE clusters and OCx clusters have switched positions between these two plots (OCx aligns closer to DPE in Fig 1c, consistent with note made in supplementary table 1). Is this by mistake or is it by chance due to similarities between these clusters? "

Response: The orientation of tSNE depends on multiple factors. The proximity of cell populations represents some similarities between clusters however, in this case, the orientation of the tSNE is mostly affected by the memory allocated to generate the analysis. Even though the tSNE in Fig 1c and 4a are oriented left-to-right and right-to-left respectively, we changed the memory allocated so that both tSNEs look as similar as possible in the manuscript. For instance, in both cases OCx is at the top and PR are at the bottom of tSNE.

DPE and OCx switched positions for similar reasons as well as because the tSNE reduces dimensionality of 20 principal components (20 dimensions) into a two-dimensional plot. For example, in a 3D tSNE plot the clusters would look completely different. However, all previous single-cell papers use a 2D tSNE for simplicity so that the plot is easier to read.

Point 3. *"Left: Figure 3a, Right Figure 1c."*

Response: We are not sure what the comment "Left 3a, Right Figure 1c" meant. We assume it relates to colors and orientation of the tSNE that we addressed in point 1 and 2.

Point 4. *"Figure 2f: What are component 1 and 2? Please define these if possible."*

Response: The "components" in the single cell trajectory are the two principal components automatically generated by the 'Monocle 2' script using Reversed Graph Embedding to reduce dimensionality, similar to tSNE. The purpose is to assign and group cells in order to visualize the temporal order of populations. In our case, we applied this script to determine whether the single cell populations from tSNE are grouped in the pseudotime as we would expect based on prior knowledge of eye disc development, for example photoreceptor cells would arise after MF. To simplify the plot, we replaced the label on the axes and added the title "Pseudotime Trajectory" (**Fig. 3f**).

Point 5. *"Figure 3b. I think an outline of approximate cluster borders would be useful. "*

Response: We attempted to outline all the clusters in the feature plots in **Figure 4b**, however, they became visually crowded and difficult to read. For this reason, we decided to circle only Cluster 13 in order to highlight the location of this cluster based on the tSNE. We hope that the reviewer agrees.

Point 6. *"I noticed that there were a lot of other interesting changes seen in the Rbf mutants, such as decreased amounts of photoreceptor cells. Are there also increased amounts of pedipodial epithelium cells? Please comment on the other mutant-specific changes observed. "*

Response: The reviewer made an important point about comparing of number of wild type and Rbf mutant cells that contribute to each cluster. Particularly, the reviewer asked whether the number of photoreceptors is lower in Rbf mutants. This was actually what we wanted to do

when we generated the Rbf mutant dataset. However, in this case this analysis proved to be uninformative. This primarily because the morphogenetic furrow is continuously moving during the eye development and separates undifferentiated cells in the anterior from differentiating photoreceptors. Thus, the number of photoreceptors is defined by the current position of the morphogenetic furrow. Since Rbf mutants exhibit a slight development delay, the MF slightly lags behind and therefore the number of photoreceptors appear to be lower than in the wild type. We mentioned this on page 12.

The reviewer also asked about possible differences in number of peripodial membrane cells in scRNA-seq datasets. To address this, we stained wild type and *Rbf* mutant eye discs with DAPI and counted over 40,000 cells (>20,000 Rbf and >20,000 WT) across 9 *Rbf^{fz08}* and 11 wild type eye discs. The results show there is a 1.26-fold increase in the number of peripodial cells in *Rbf* relative to WT. However, we performed an ANOVA test of significance which showed this increase not to be significant, $p = 0.18$. Since the differences are not statistically significant we decided not to include it in the manuscript.

Point 7. “Figure 4B. *Ald*, *HIF1a* and *Ldh* RNAi seem to reduce apoptosis, but there is still a lot of apoptosis going on. Could this be due to efficiency of the RNAi's, or is apoptosis affected by reduced metabolism as a consequence of reduced glycolytic activity? Can apoptosis be induced by overexpression of *Ldh*, *HIF1a* or *Ldh* in non-mutant cells?”

Response: We agree with the reviewer that incomplete suppression of apoptosis following depletion of *Ald*, *HIF1A*, and *Ldh* RNAi is likely due to the efficiency of RNAi *in vivo*. To address the question of inducing apoptosis (and low pH), we performed overexpression experiments for *HIF1a* using *ey-Gal4* and *eyFLP*, *Act5c-FRT-stop-FRT-Gal4* drivers. These resulted in the complete loss of the eye disc at this developmental stage. Therefore, we decided to overexpress *HIF1a* using a *heatshock-Gal4* driver, which would be activated during a short treatment (10, 30, 60 minutes) at 37°C. We then immunostained for cleaved Dcp-1 and pH to study cell death and cellular acidification, respectively, however, the results were not informative and were not included.

Point 8. “Figure 4C. *E2F1* seems to interact weakly with *Ald* and *HIF1a* compared to *Rbf*, *Dp* and *E2F2*. What larval tissues have been used? Cycling or quiescent tissue? Are the differences measured significantly different (p values and which statistical test)? Do error bars represent Standard deviation or SEM?”

Response: The reviewer is correct that E2f1 ChIP signal is weaker than Rbf, E2f2 or Dp. This is a well-known property of E2f1 antibody that we and others observed. The level of enrichment of E2f1 genomic occupancy is usually not very high even in the promoters of well-known E2f1-target genes. For example, see: Korenjak et al (2012) MCB, Ambrus et al (2013) Dev Cell, Nicolay et al (2011) Genes Dev, Moon et al (2005) Dev Cell, Dimova et al (2003) Genes Dev, and Zappia and Frolov (2016) Nat Commun. However, it is unreliable to compare the relative enrichment of ChIP signal using different antibodies. Thus, we cannot conclude that E2f1 binds to promoter with less affinity than RBf or E2f2.

Whole third instar larvae were used to prepare chromatin. This information is now added to the manuscript on page 14. At this developmental point, larva contain both cycling, endocycling and quiescent tissues.

The differences measured are significantly different. The error bars represent Standard deviation. The details on statistical analysis information are now included in the legend on page 38.

Point 9. *“Figure 4D and discussion. They claim lower pH results in apoptosis. They should artificially increase the pH (through Ldh over-expression?), and look for increased apoptosis in the cells with low EGFR activity.”*

Response: As described in our response to point 7, we attempted overexpressing of HIF1a since it is known to directly regulate Ldh expression but the results were inconclusive due to high degree of lethality. We have also discussed whether the intracellular acidification arises independently of the activation of pro-apoptotic genes on page 14 and added the experiment that shows that *hid* overexpression is insufficient to lower pH (**Supplementary Fig. 3**). Inclusion of *hid* on page 14 is important since it is a direct target of E2F, and another marker of Cluster 13. It has also been previously shown that apoptosis in *Rbf* mutants is driven through E2F-activation of *hid* (Moon, N-S., et al, 2006).

Reviewers' Comments:

Reviewer #1:

Remarks to the Author:

The reviewers have addressed all points raised. In particular the addition of figure 2 is helpful to use the data of the current manuscript as resource.

As minor comment to the lack of Scenic as analysis tool I feel it would be beneficial to include different analysis, however I do understand the issue stated by the authors. Overall the adaptations of the manuscript make it a strong contribution to the field

Reviewer #2:

Remarks to the Author:

I am satisfied with the author's responses. I would only request that the author's submit their data to the Broad Institute's Single Cell Portal (can keep the study private until acceptance) to enable easy visualization the underlying data.

https://portals.broadinstitute.org/single_cell

Reviewer #3:

Remarks to the Author:

This manuscript has undergone substantial revisions in response to the reviews. One of the concerns expressed by the reviewers was that the paper should present the data in such a way that the eye atlas can be used as a "cornerstone" for eye-disc studies. The authors have responded by providing more information on eyedisc markers and a new figure 2 was produced as a result. It's more comprehensive than before, and the fact that this data has been submitted to GEO strengthen this publication as a valuable resource. Another question raised addressed the shortcomings of the Rb study, which was expanded upon by the authors. Another good point raised by reviewer #2 was that that the authors claimed that SMW cells were enriched in mitotic markers, but provided no data to support the claim. This was corrected satisfactorily by the authors. The same reviewer requested a full heatmap in figure 1e. The authors expanded this figure, improving the presentation, but it seems they've unintentionally left out labels for the clusters along one axis. These were displayed in the original figure 1e, and need to be restored here in the revised figure to make it useful. Another reviewer asked whether the authors could test in a more direct manner their suggestion that intracellular acidification is sufficient to induce apoptosis in WT cells. The authors explained that attempts to do this produced inconclusive data. Thus, this issue may be considered minor weakness in their proposal that intracellular acidification increases apoptosis. Thus weakness should be mentioned in the discussion. Nevertheless the data on acidification causing death in Rbf mutant cells is convincing. Overall the paper is good quality work that presents a big dataset on an interesting topic. It provides a great technical example of the efficacy of scRNAseq, and reveals an aspect of Rbf control that's unique. How important the cellular acidification caused by Rbf loss is in other contexts, like human cancer, is not yet clear, but the possibility is interesting. This work certainly warrants publication.

Reviewer #1:

The reviewers have addressed all points raised. In particular the addition of figure 2 is helpful to use the data of the current manuscript as resource.

As minor comment to the lack of Scenic as analysis tool I feel it would be beneficial to include different analysis, however I do understand the issue stated by the authors. Overall the adaptations of the manuscript make it a strong contribution to the field

Response:

Nothing is requested.

Reviewer #2:

I am satisfied with the author's responses. I would only request that the author's submit their data to the Broad Institute's Single Cell Portal (can keep the study private until acceptance) to enable easy visualization the underlying data.

https://portals.broadinstitute.org/single_cell

Response:

We submitted the scRNA-seq data to the Broad Institute Single Cell Portal. The data are publicly available now.

Reviewer #3:

This manuscript has undergone substantial revisions in response to the reviews. One of the concerns expressed by the reviewers was that the paper should present the data in such a way that the eye atlas can be used as a "cornerstone" for eye-disc studies. The authors have responded by providing more information on eyedisc markers and a new figure 2 was produced as a result. It's more comprehensive than before, and the fact that this data has been submitted to GEO strengthen this publication as a valuable resource. Another question raised addressed the shortcomings of the Rb study, which was expanded upon by the authors. Another good point raised by reviewer #2 was that that the authors claimed that SMW cells were enriched in mitotic markers, but provided no data to support the claim. This was corrected satisfactorily by the authors. The same reviewer requested a full heatmap in figure 1e. The authors expanded this figure, improving the presentation, but it seems they've unintentionally left out labels for the clusters along one axis. These were displayed in the original figure 1e, and need to be restored here in the revised figure to make it useful. Another reviewer asked whether the authors could test in a more direct manner their suggestion that intracellular acidification is sufficient to induce apoptosis in WT cells. The authors explained that attempts to do this produced inconclusive data. Thus, this issue may be considered minor weakness in their proposal that intracellular acidification increases apoptosis. Thus weakness should be mentioned in the discussion. Nevertheless the data on acidification causing death in Rbf mutant cells is

convincing. Overall the paper is good quality work that presents a big dataset on an interesting topic. It provides a great technical example of the efficacy of scRNAseq, and reveals an aspect of Rbf control that's unique. How important the cellular acidification caused by Rbf loss is in other contexts, like human cancer, is not yet clear, but the possibility is interesting. This work certainly warrants publication.

Response:

The reviewer asks about labeling of axis on Figure 1e that shows a heat map. Cell populations are labeled on X-axis, while genes are plotted along Y-axis. Given the large number of genes for each cell population and low magnification we decided not to list them as it will make the figure look very busy. This is a common practice and is used in other papers on scRNA-seq.

As requested, we modified Discussion to acknowledge that we did not show that increased expression of glycolytic genes induces apoptosis in wild type eye discs.